# Multisensor monitoring and data integration reveal cyclical destabilization of Äußeres Hochebenkar Rock Glacier

Lea Hartl[1, 6], Thomas Zieher[1], Magnus Bremer[1, 2], Martin Stocker-Waldhuber[1], Vivien Zahs[3], Bernhard Höfle[3, 4], Christoph Klug[2], and Alessandro Cicoira[5]

[1] Institute for Interdisciplinary Mountain Research, Austrian Academy of Sciences, Innrain 25, 3. OG 6020 Innsbruck, Austria
[2] Department of Geography, University of Innsbruck, Innrain 52f, 6020 Innsbruck, Austria
[3] Institute of Geography, Heidelberg University, Im Neuenheimer Feld 368, 69120 Heidelberg, Germany
[4] Interdisciplinary Center for Scientific Computing (IWR), Heidelberg University, Im Neuenheimer Feld 205, 69120 Heidelberg, Germany
[5] Department of Geography, University of Zurich, Winterthurerstr. 190, 8057 Zurich, Switzerland
[6] Alaska Climate Research Center, Geophysical Institute, University of Alaska Fairbanks, 2156 Koyukuk Dr, Fairbanks, AK 99775, United States

**Correspondence:** Lea Hartl (Lea.Hartl@oeaw.ac.at)

**Abstract.** This study investigates rock glacier destabilization based on the results of a unique in situ and remote sensing-based monitoring network focused on the kinematics of the rock glacier in Äußeres Hochebenkar (Austrian Alps). We consolidate, homogenise, and extend existing time series to generate a comprehensive dataset consisting of 14 digital surface models covering a 68 year time period, as well as in situ measurements of block displacement since the early 1950s. The digital surface models are derived from historical aerial imagery and, more recently, airborne and uncrewed aerial vehicle-based laser scanning (ALS, ULS). High-resolution 3D ALS and ULS point clouds are available at annual temporal resolution from 2017 to 2021. Additional terrestrial laser scanning data collected in bi-weekly intervals during the summer of 2019 is available from the rock glacier front. Using image correlation techniques, we derive velocity vectors from the digital surface models, thereby adding rock glacier-wide spatial context to the point scale block displacement measurements. Based on velocities, surface elevation change, analysis of morphological features, and computations of the bulk creep factor and strain rates, we assess the combined datasets in terms of rock glacier destabilization. To additionally investigate potential rotational components of the movement of the destabilized section of the rock glacier, we integrate in situ data of block displacement with ULS point clouds and compute changes in the rotation angles of single blocks during recent years. The time series shows two cycles of destabilization in the lower section of the rock glacier. The first lasted from the early 1950s until the mid 1970s. The second began around 2017 after approximately two decades of more gradual acceleration and is currently ongoing. Both destabilization periods are characterized by high velocities and the development of morphological destabilization features on the rock glacier surface. Acceleration in the most recent years has been very pronounced, with velocities reaching 20-30 m/a in 2020/21. These values are unprecedented in the time series and suggest highly destabilized conditions in the lower section of the rock glacier, which shows signs of translational as well as rotational, landslide-like movement. Due to the length and granularity of the time series, the cyclic destabilization process at Äußeres Hochebenkar rock glacier is well resolved in the dataset. Our study

highlights the importance of interdisciplinary, long-term and continuous, high-resolution 3D monitoring to improve process understanding and model development related to rock glacier rheology and destabilization.

## 1 Introduction

Rock glaciers are lobate or tongue shaped landforms supersaturated by ice and generated by the former or current creep of frozen ground (Barsch, 1992; Haeberli et al., 2006; RGIK, 2022). They are widespread features of the mountain cryosphere and serve as proxies for former and current permafrost occurrence, play an important role in the hydrological system, and form part of the sediment cascade (Wahrhaftig and Cox, 1959; Barsch, 1996; Haeberli et al., 2006; RGIK, 2022). Rock glacier kinematics have increasingly become of interest during approximately the last two decades due to accelerated movement at many rock glaciers in the Alps and in creeping subarctic permafrost (Roer, 2005; Delaloye et al., 2008, 2010; Daanen et al., 2012; Kellerer-Pirklbauer et al., 2018; Fleischer et al., 2021). These general trends are broadly attributed to rising air and ground temperatures. However, interannual variations of rock glacier velocities often cannot be explained with variations in mean annual or summer air temperatures alone. This indicates a multifaceted and scale-dependent relationship between climate forcing and rock glacier response (Delaloye et al., 2008, 2010; Sorg et al., 2015; Hartl et al., 2016b; Kellerer-Pirklbauer et al., 2017; Müller et al., 2016; Cicoira et al., 2019a; Fleischer et al., 2021). Sub-seasonal monitoring of rock glacier movement has shown correlations between short-term velocity fluctuations and water input from snow melt or precipitation. Liquid water within the rock glacier strongly affects the material properties relevant to the rheology and, hence, kinematics of the moving mass (Krainer and He, 2006; Wirz et al., 2016; Kenner et al., 2017; Buchli et al., 2018; Cicoira et al., 2019b; Fleischer et al., 2021).

In recent years, the term *destabilization* has gained traction in the context of rock glacier acceleration. While there is no universal, exact definition of rock glacier destabilization to date, it is generally understood to mean anomalous "landslide-like behaviour" (Marcer et al., 2021). Destabilization is characterized by a sudden, strong, often localized increase in velocity and the concurrent appearance of morphological destabilization signs like cracks, crevasses, and scarps (Roer, 2005; Roer et al., 2008; Delaloye et al., 2013). Destabilization appears to require favourable terrain, i.e., a relatively steep slope angle, and onset often occurs at convexities or terrain steps (Marcer et al., 2019, 2021). Velocity discontinuities and morphological destabilization signs develop in these areas and the rock glacier is essentially split into a destabilized (usually lower) section and a comparatively unaffected (usually upper) section (Cicoira et al., 2020). Destabilization features like crevasses and deep cracks allow rain or melt water to enter the rock glacier, further contributing to acceleration and changes in rheology and kinematics (Roer, 2005; Delaloye et al., 2013; Buchli et al., 2018; Eriksen et al., 2018; Vivero and Lambiel, 2019). In terms of rheology, it is assumed that sliding on shear horizons, which may be basal or located within the structure of the rock glacier, dominates the destabilized movement. In contrast, "normal" rock glacier movement is driven by viscous creep of the ice-rich permafrost core (Arenson et al., 2002; Roer et al., 2008; Krainer et al., 2015; Schoeneich et al., 2015; Marcer et al., 2019; Cicoira et al., 2019b, 2020). The underlying causes of destabilization may be climatic (rheological changes due to increased temperature and/or liquid water input, e.g., Delaloye et al. (2013)), mechanical (local overloading due to rock fall events and subsequent

propagation of compressive processes, e.g., Scotti et al. (2017)), or a combination thereof. We follow the terminology of Marcer et al. (2021) regarding rock glacier destabilization and refer to their publication for a more comprehensive overview of the conceptual framework around different phases of destabilization. In the following, we use the term "destabilization signs" to refer to visible morphological features such as scarps, crevasses, and cracks that develop at the onset of and during destabilization.

Relatively little is known about regional distribution of destabilized rock glaciers. Marcer et al. (2019) find evidence of destabilization at 10% of active rock glaciers in France based on topographic data from 2000 to 2013. Interestingly, they note that destabilizing rock glaciers tend to be pebbly as opposed to bouldery (Ikeda and Matsuoka, 2006) and located in densely jointed lithologies (i.e. ophiolites and schists) as opposed to crystalline lithologies. Expanding their analysis further into the past, Marcer et al. (2021) identify a small number of rock glaciers in France that experienced a form of destabilization around the middle of the 20th century. These rock glaciers then returned to normal behaviour and began destabilizing once again in the last two to three decades. Case studies have detailed the destabilization of rock glaciers throughout the Alps in Switzerland, France, Italy, and Austria (Avian et al., 2005; Delaloye et al., 2013; Scotti et al., 2017; Vivero and Lambiel, 2019; Kaufmann et al., 2021; Bearzot et al., 2022; Vivero et al., 2022). While not all accelerating rock glaciers are necessarily destabilized, pronounced or unusual acceleration patterns may hint at destabilization processes at the respective sites. This applies to the Alps as well as other mountain regions, even though related studies do not always explicitly use the term destabilization to describe the observed changes (Delaloye et al., 2010; Daanen et al., 2012; Hartl et al., 2016b; Eriksen et al., 2018; Kääb et al., 2021).

In most cases, destabilization is followed by degradation (Cicoira et al., 2020), i.e., deceleration and eventual inactivity of the previously destabilized section of the rock glacier. In rare cases, the complete collapse of the destabilized section followed by a rapid mass movement of substantial parts of the affected rock glacier has been observed. Such events tend to be preceded by high temperatures and wet conditions caused by, e.g., significant individual precipitation events, longer periods with anomalous amounts of precipitation, or water input through snow melt. Morphological destabilization signs prior to the collapse have been reported in cases for which detailed analyses of collapse events exist (Krysiecki et al., 2008; Bodin et al., 2012, 2017; Marcer et al., 2020; Kofler et al., 2021). At some sites, e.g., Grabengufer rock glacier in Switzerland, the combination of very high velocities, obvious signs of destabilization, and downstream topography favourable for mass movements led to concern but did not produce debris flows (Delaloye et al., 2013). This suggests that the underlying processes are complex and depend on a variety of internal (composition of the rock glacier, internal hydrological processes) and external (meteorological and climatological forcing, surrounding terrain) factors (Marcer et al., 2019, 2021).

## 1.1 Äußeres Hochebenkar rock glacier

In the following, we present data from an interdisciplinary monitoring network at the rock glacier in Äußeres Hochebenkar (HEK) in the Austrian Alps (Fig. 1) and discuss the behaviour of the lower section of the rock glacier through the lens of destabilization. HEK is a fast moving, tongue-shaped talus rock glacier with a long history of scientific study (Table 1). It is one of 556 intact rock glaciers in the Ötztal mountain range (Wagner et al., 2020) and covers an area of about 0.4 km$^2$. The

root zone reaches a maximum elevation of about 2870 m and the terminus currently extends down to about 2370 m. (Note: Unless otherwise specified, all elevation values in this manuscript refer to orthometric heights; EVRF2000 Austria height, EPSG:9274.) The slope angle of the rock glacier is moderate in the upper section and steepens below a terrain step at around 2570 m. The terminus funnels into a drainage gully, which is located directly above an access road leading to mountain huts further up-valley. The road has recently been affected by rock fall from the rock glacier terminus and is threatened by any further destabilization.

Climatologically, HEK is located in a relatively dry, inner-alpine valley. The mean annual air temperature (MAAT) in nearby Obergurgl (1938 m.a.s.l.) was 2.2 °C during the 1961-1990 reference period and 2.7°C during the 1991-2020 reference period (Kuhn et al., 2013). Mean annual precipitation sums at the Obergurgl weather station were between about 840 mm and 900 mm depending on the reference period (data obtained from ZAMG data hub, data.hub.zamg.ac.at, Dautz et al. (2022)). Meteorological data from an automatic weather station at HEK (2565 m.a.s.l.) shows that MAAT at the rock glacier was close to 0°C in recent years (Stocker-Waldhuber et al., 2013).

Using refraction seismics, Haeberli and Patzelt (1982) found the mean thickness of the rock glacier to be about 40 m and estimated an ice content of 50%. More recent ground penetrating radar (GPR) surveys found comparable values for the mean depth of the bedrock, with considerable variation in depth throughout the rock glacier area, and a several metres (up to >10 m) thick surface layer of ice-free, coarse debris (Nickus et al., 2015; Hartl et al., 2016a). It should be noted that the depth of the bedrock is not necessarily the same as the thickness of the moving mass or the thickness of the thermally defined permafrost within the rock glacier. The surface debris has an average grain size of 35 cm to about 60 cm, with some blocks reaching diameters of up to a few metres (Nickus et al., 2015). A layer of finer material below the coarse surface debris is exposed at the steep frontal slope of the rock glacier. The surrounding lithology is part of the crystalline Ötztal-Stubai complex and the bedrock is composed mainly of paragneiss and mica shists (Nickus et al., 2015).

Surveys of the temperature at the bottom of the winter snow cover (BTS) carried out in 1976 (Haeberli and Patzelt, 1982) and 2010 indicate a reduction in the extent of the discontinuous permafrost surrounding the rock glacier during this period, particularly at the margins of the rock glacier and directly below the terminus (Nickus et al., 2015). This is in line with the overall degradation of ground ice in the Alps as reported by, e.g., Etzelmüller et al. (2020). The rock glacier terminus was at the lower end of the discontinuous permafrost margin even in the 1970s and has advanced downwards since, hence moving further below the likely permafrost margin. The permafrost index map of Boeckli et al. (2012) suggests that permafrost is relatively likely in the upper parts of HEK, but likely only under cold or very favourable conditions near the terminus. Zahs et al. (2019) found no permafrost in an electrical resistivity tomography (ERT) profile beside the rock glacier at an elevation of about 2470 m. In a profile at similar elevation on the rock glacier terminus they reported resistivities indicative of ice-rich frozen ground with variable ice content.

Regular measurements of block displacements on the surface of HEK rock glacier began in the early 1950s and continue to this day. Since the 2000s, numerous remote sensing-based studies have used a variety of methods and analysis techniques to quantify surface elevation change and horizontal displacement at HEK, providing spatial context to the in situ monitoring of block movement. Summarising broadly, previous studies show that vertical surface elevation changes are typically small or

slightly negative in the upper part of the rock glacier, while the lower section is extending downwards and thinning. See Table 1 for an overview of the respective literature.

Morphological observations in early studies and the long time series of block velocities indicate processes of destabilization at HEK in the 1950s and '60s: During surveys in the 1970s, Haeberli and Patzelt (1982) observed large crevasses in the lowest section of the rock glacier tongue. They noted one crevasse crossing the entire width of the rock glacier at around 2540 m and speculated that the lowest section of the terminus below this crevasse had previously undergone a process of separation or decoupling from the upper part. Schneider and Schneider (2001) interpreted the kinematics of the lower section of the rock
glacier in a similar manner, agreeing with the idea that the terminus area had separated from the main body of the rock glacier. They described the main part as being in a normal state ("*Normalzustand*") and healthy ("*gesund*"), whereas the behaviour of the presumably separated part was considered extraordinary ("*außergewöhnlich*"). Schneider and Schneider (2001) further hypothesised that the process of separation of the entire section below the terrain step was completed by the early 1970s, and that it might repeat in the future given continued advancement of the rock glacier over the terrain step. They speculated that
the lowest section of the terminus became inactive after this destabilization phase due to the loss of ice-rich permafrost in the terminus area. Schneider and Schneider (2001) attributed the anomalous behaviour in the lower section to the steep slope angle and basal sliding processes as opposed to the permafrost creep governing the upper section, describing the same general destabilization processes as more recent work (Roer et al., 2008; Schoeneich et al., 2015; Marcer et al., 2019; Cicoira et al., 2020).

After this early period of high velocities and destabilization, the rock glacier returned to its "normal" state until the onset of a second phase of acceleration in the mid-1990s. The lower section below the terrain step and the crevasse observed by Haeberli and Patzelt (1982) also accelerated again with a peak in velocities in 2004 and a stronger, currently ongoing acceleration after 2010 (Hartl et al., 2016b).

## 1.2 Objectives

Our overarching goal is to contribute to the developing scientific discussion around rock glacier destabilization by consolidating and making available the in situ and remote sensing data basis from our study site. We provide a comprehensive overview of two separate destabilization periods at the same rock glacier by combining the most recent multi-sensor and multi-method monitoring results with data from previous studies and long-term observations (Table 1). Specifically, we:

– Homogenise and update a time series of 14 digital surface models (DSMs) derived from aerial imagery (1953-1997,
Klug (2011)), and LiDAR point clouds, also referred to as laser scanning (2006-2021).

– Present the most recent data from the long-term in situ time series of DGNSS-based block displacement, updating the dataset presented in Hartl et al. (2016b) (time series: 1952-2022).

– Compute a time series of the bulk creep factor (BCF) as a metric of destabilization (Cicoira et al., 2020) to aid interpretation of velocity change

– Extract additional information about the rock glacier's recent surface change from very high-resolution (cm point spacing) 3D point clouds. We derive (a) 3D change information between two epochs based on corresponding planar boulder faces at the rock glacier front (Zahs et al., 2022a, b) and (b) a time series of rotating single blocks.

    Following an overview of the different datasets and methods, the results are structured chronologically, moving from the first period of destabilization to the subsequent "stable" period and the start of the ongoing renewed acceleration and destabilization.
For each period, we describe kinematic and morphological changes based on the in situ and remote sensing datasets. For the most recent years, we present additional results based on the fusion of high-resolution laser scanning data and block displacement, as well as data on sub-seasonal 3D topographic change at the rock glacier front for the summer of 2019. In the discussion, we then consider uncertainties and limitations related to the different datasets and methods, and offer an interpretation of the observed changes in the general context of rock glacier destabilization.

Table 1: Overview of previous studies at HEK rock glacier, grouped by thematic focus and listed chronologically per thematic group.

**Interdisciplinary overview publications**

| | |
|---|---|
| Permafrost mapping based on BTS survey, descriptions of morphological features on the rock glacier surface, refraction seismics. | Haeberli and Patzelt (1982) |
| Geological setting, BTS data, presentation of first GPR survey, stream flow data and analysis of conductivity and chemical composition of rock glacier runoff. | Nickus et al. (2015) |

**Displacement of block profiles**

| | |
|---|---|
| Details on establishment of block profiles and first 2 decades of measurements. | Vietoris (1958, 1972); Pillewizer (1957) |
| Digitisation of early data, homogenisation of time series, systematic assessment of mean profile velocities and single blocks, analysis of surface elevation change at the block locations. | Schneider (1999a, b) |
| Overview of block movement since beginning of measurements and consolidation of results of Schneider (1999a, b); considerations on role of climate parameters as drivers of block movement. | Schneider and Schneider (2001) |
| Update to time series of block profiles including profile 0 (established in 1997). Statistical analysis of correlation between cumulative anomalies of movement and annual and seasonal means of climate parameters. | Hartl et al. (2016b) |

**Remote sensing**

| | |
|---|---|
| Terrestrial photogrammetric surveys of the rock glacier tongue (Advance of 1.1 m between 1953-55 at the main, orographic left lobe); No movement detected at the smaller, orographic right lobe for 1956-66 (unpublished data cited in Vietoris (1972)) | Vietoris (1972) |
| Terrestrial photogrammetric surveys of the lower section of the rock glacier in 1986, 1999, 2003, 2008, and detailed analysis thereof. Computation of 3D flow vectors. Overview and analysis of historic cartographic and photogrammetric data. | Kaufmann and Ladstädter (2002b, a); Ladstädter and Kaufmann (2005); Kaufmann (2012) |
| Analysis of orthophotos to generate digital surface models (DSM) and horizontal flow vectors (orthophotos available for: 1953, 1969, 1971, 1977, 1990, 1997) | Klug (2011); Klug et al. (2012) |
| Analysis of airborne laser scanning (ALS) data to generate DEM and flow vectors (ALS data for 2006, 2009, 2010, 2011) | Bollmann et al. (2012); Klug et al. (2012) |
| Multi-source (ALS, terrestrial laser scanning (TLS), UAV-borne laser scanning (ULS), and UAV-borne photogrammetry-based dense image matching (DIM)) 3D topographic change analysis for different time spans in the period 2006-2021, including bi-weekly monitoring in summer 2019. | Zahs et al. (2019); Ulrich et al. (2021); Zahs et al. (2022b, a) |

**Geophysics**

| | |
|---|---|
| Refraction seismics: Surveys carried out in 1975 and 1977. Mean ice content of about 50% at surveyed profiles. | Haeberli and Patzelt (1982) |
| Ground penetrating radar: Surveys carried out in 2000, 2008, 2013 with two different radar systems at different locations on the rock glacier. Mean depth of bedrock between 34 and 45m depending on survey year. | Nickus et al. (2015); Hartl et al. (2016a) |
| Electrical resistivity tomography: Survey carried out in 2016 at two profile lines next to and on the margin of the rock glacier tongue. No indications of permafrost beside the rock glacier, isolated ice lenses identified in the profile on the tongue. | Zahs et al. (2019) |

## 2 Data and methods

### 2.1 Long-term geodetic surface displacement monitoring

Since 1954, surface displacement of HEK rock glacier has been measured geodetically along three cross profiles (Vietoris, 1958, 1972). See Fig.1 for an overview of the profile locations. Measurements were initially carried out annually but the time series has substantial gaps from the 1960s until 1997, when a fourth cross profile and a longitudinal profile were installed in the lowest section and the monitoring efforts were revitalized (Fig. 2). Schneider (1999a, b); Schneider and Schneider

(2001), and Niederwald (2009) give details on the homogenisation of the time series and early measurement techniques. Since 2008, displacement measurements have been carried out with a Topcon HiperPro real-time kinematic differential global navigation satellite system (DGNSS), replacing a theodolite and tachymeter previously used for the same purpose. Hartl et al. (2016b) describe the current measurement system and give an overview of the kinematics of HEK rock glacier up to 2015.

Historically, reporting focused on mean displacement values for the cross profiles P0, P1, P2, and P3. The mean value was computed as the arithmetic mean of all blocks in the profile and displacement refers to the absolute, three-dimensional distance moved. Given issues related to averaging and changing numbers of blocks in the profiles, Hartl et al. (2016b) estimate an uncertainty of between $\pm 0.2$ m/a and $\pm 0.5$ m/a for the mean profile velocities. We adhere to the same method of using 3D trajectories of single blocks as part of a profile when referring to profile means in order to be consistent with the long-term time
series. Measurements are carried out annually in summer, usually in late August or early September. Annual mean values are derived from the absolute displacement and the number of days between measurement campaigns. The tachymeter and DGNSS displacement measurements for single blocks are considered accurate to $\pm 3$ cm vertically and horizontally (Niederwald, 2009; Nickus et al., 2015). In the 2021/22 measurement year, multiple marked blocks in P1 were lost due to the rapid movement of the rock glacier in this area. We present velocities for the remaining blocks in the profile for the most recent measurement
year but do not compute a mean profile velocity, as this would no longer be representative given the lost blocks. We refer to the existing literature for further details on the time series of block displacement. (See Table 1; in particular Schneider and Schneider (2001); Niederwald (2009); Hartl et al. (2016b)).

The dynamics of destabilized rock glaciers are characterized by velocity discontinuities between the faster, destabilized (usually lower) section and the slower, non-destabilized (usually upper) section. As this discontinuity becomes more pronounced
under ongoing destabilization, the surface strain between the two sections also increases (Marcer et al., 2021). Hence, changes in surface strain rates can serve as indicators of destabilization onset in specific sections of a rock glacier. To quantify these changes for recent years at HEK, we use the positions of individual blocks in P1 and P2 to compute surface strain rates across the terrain step and the velocity discontinuity located between P1 and P2, i.e., between the currently destabilized lower section and the non-destabilized upper section. The surface strain rate between a pair of blocks (b1, b2) in P1 and P2 is given by the
difference in velocity between b1 and b2 divided by the distance between b1 and b2, following Marcer et al. (2021).

## 2.2 Meteorological data

As part of the HEK monitoring network, an automatic weather station (AWS) was installed at 2565 m in 2010 directly beside the rock glacier (Stocker-Waldhuber et al., 2013; Hartl and Fischer, 2015). Long-term meteorological data is available from a semi-automatic weather station in Obergurgl since 1953. This station is located about 4 km down-valley from the rock
glacier. From 1953 to 1998 the Obergurgl station was located at 1938 m.a.s.l. It was then moved to a nearby location at 1941 m.a.s.l. Data is available from the data portal of the Austrian Central Institution for Meteorology and Geodynamics (ZAMG, https://data.hub.zamg.ac.at/), which operates the station (Dautz et al., 2022). An overview of the meteorological time series from the Obergurgl station can be found in Kuhn et al. (2013).

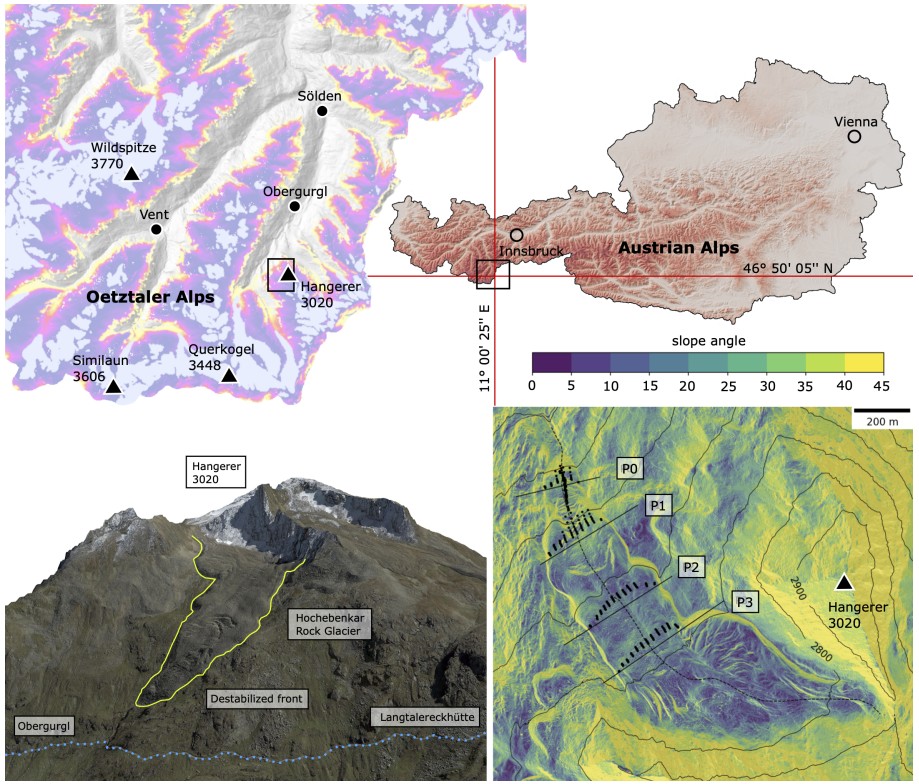

**Figure 1.** Location of the study area in the Austrian Alps (top right, base data: NASA Shuttle Radar Topography Mission (SRTM), red shading: elevation) and detail of the geographical context of the Hochebenkar Rock Glacier in the Ötztal Alps (top left, base data: SRTM, the overlay shows the permafrost index map as per Boeckli et al. (2012). Light blue color: Glacier; dark blue: Permafrost in nearly all conditions; purple: Permafrost mostly in cold conditions; yellow: Permafrost only in very favourable conditions). 3D visualization of the study site (data from SRTM and GoogleEarth, bottom left) and slope angle map draped over hillshade with contour lines, all generated from the 2017 DSM (bottom right). Black dots mark the positions of the blocks in the four cross profiles and the longitudinal profile for 2015-2021. Also shown are the central flowline (dashed black line) and the lines between the fixed reference points that define the positions of the cross profiles (solid black lines).

## 2.3 Remote-sensing-based area-wide monitoring of topographic change

Table 2 and Fig. 2 give an overview of all topographic data used for this study, which includes 14 DSMs covering a time period of 68 years, as well as orthophotos from UAV surveys and TLS-based point clouds of the front and lower terminus area. Photogrammetrically reconstructed surface topography based on historical analogue aerial imagery is available for multiple years between 1953 and 1997. Please see Klug (2011) for details on the aerial imagery and related processing steps. In more recent years, various airborne (ALS), terrestrial (TLS) and UAV-based laser scanning (ULS) surveys were carried out at HEK.

We describe the ULS data acquisition set up and the give details on the processing of the TLS datasets in section 2.3.1. In section 2.3.2, we describe the work flow used to create a homogenised time series of DSMs that combines the photogrammetrically

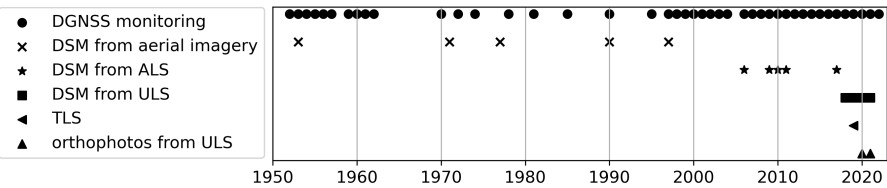

**Figure 2.** Overview of data sets used in this study: DSMs, orthophotos and TLS data as detailed in Table 2, DGNSS monitoring refers to in situ, geodetic monitoring of block displacement.

reconstructed surface topography (Klug, 2011) and the recent ALS and ULS datasets. We further describe time series of surface velocity and elevation change that were derived from the DSMs. Finally, in section 2.3.3, we describe how rotational information for individual blocks was computed from a combination of ULS point clouds and results of the DGNSS monitoring.

### 215 2.3.1 Recent high-resolution monitoring of 3D topographic change

In 2018, 2019, 2020 and 2021, ULS campaigns were conducted with a Riegl RiCopter UAV, carrying a VUX-1LR kinematic laser scanner, run with a 336° field of view and combined with an Applanix AP20 inertial measurement unit (IMU) (Bremer et al., 2019). Additionally, a Nadir-looking Sony Alpha 6000 RGB camera was mounted for orthophoto creation. In 2018, only the lower terminus area was captured. The following flight campaigns were extended to the middle part of the rock glacier.
All acquisitions followed the same flight plan: Aside from small connection lines, the rock glacier was captured by a set of parallel flight lines with a horizontal spacing of 100 m, oriented perpendicular to the flow direction of the rock glacier. The average flying height above ground level (AGL) was between 70 m and 120 m. The flight speed was 8 m/s, the pulse repetition rate (PRR) was 820 kHz and the angular resolution was 0.0476°. Following standard procedures such as flight trajectory post-processing, point extraction, geo-referencing and strip-adjustment, the resulting point clouds were co-registered to the ALS
2017 dataset by using the ICP algorithm to minimize distances between the point clouds in stable areas of bedrock outcrops (For details on the 2017 dataset used as reference, please see Table 2 and Rieger (2019)). In addition to the processed point clouds, 0.1 m and 1 m resolution DSMs and 3 cm resolution orthophotos were created.

To quantify sub-seasonal changes at the rock glacier front and lower terminus, a bi-weekly (June 24 to August 30) time series of six terrestrial laser scanning (TLS) point clouds was captured in 2019. This temporally highly resolved dataset complements
an annual TLS time series starting in 2015 and described in Ulrich et al. (2021) and Zahs et al. (2022a). We refer to these publications for more detailed information on the TLS data acquisition and measurement set up. Bi-weekly change in 2019 was computed using the correspondence-driven plane-based M3C2 (CD-PB M3C2) algorithm (Zahs et al., 2022b). The CD-PB M3C2 reduces uncertainty in the quantification of small-magnitude (<0.1 m) 3D topographic change. Thus, it allows confident detection of small changes in natural landscapes with complex surface dynamics, i.e. locally planar but overall rough
morphology. To derive 3D change between two epochs t1 and t2, the algorithm computes the distance between corresponding planar areas (plane pairs). Change is thereby computed along the normal vector of the plane in t1. When applied to rock

glaciers, the method can make use of the surface of a rock glacier being most planar at the scale of faces of individual boulders. These boulders move with the general creep of the rock glacier rather than independently and can can be re-identified in successive epochs due to their moderate magnitudes of movement at the monitored interval (Ulrich et al., 2021). We used these corresponding planar boulder faces for change analysis of the rock glacier front and lower terminus between the epochs. Change analysis was carried out in the flow direction of the rock glacier, in the vertical direction, and in the horizontal direction. The CD-PB M3C2 algorithm additionally estimates the uncertainty associated with quantified change. It therefore allows confident analysis of change by separating significant change (magnitude of change > uncertainty) from non-significant or no change (magnitude of change ≤ uncertainty).

### 2.3.2  Long-term change monitoring

The photogrammetrically reconstructed topography and the 3D point clouds derived from ALS and ULS were co-registered within bedrock outcrops assumed to be stable throughout the established time series. For the co-registration, the iterative closest point algorithm (ICP, Besl and McKay (1992)) was used, implemented in C++ in an extension of the SAGA GIS software (Conrad et al., 2015). The 2017 ALS data provided by the government of the state of Tyrol (Table 2) served as reference for the co-registration. The selected stable areas are distributed across the study area and include varying slope angles and orientations in order to reach a robust co-registration. Due to the topography of the cirque in which the rock glacier is located, only relatively small parts of the study area have slopes oriented mainly to the North and South and minor shifts of the registration in North-South-direction cannot be ruled out. The implications of this are discussed in Section 4.1.

After co-registration, DSMs with a spatial resolution of 1 m were computed for each epoch of the multi-temporal point clouds by aggregating the average elevation of all points per raster cell. This 1 m resolution matches the resolution of the DSMs previously derived from aerial imagery (Klug, 2011). The DSMs computed from the digitized analogue images do not include the same level of topographic detail as the DSMs derived from laser scanning. In some cases, the images have localized shading effects in the steeper sections of the terminus, which produces gaps in the topographic information extracted from the images. Please see Klug (2011) and Klug et al. (2012) for more detailed descriptions of the aerial imagery and the resulting DSMs. All datasets were projected according to the Austria GK West definition (EPSG 31254, with the vertical datum EVRF2000 Austria heights, EPSG 9274).

From the DSMs, shaded reliefs were computed with the ambient occlusion method (Tarini et al., 2006), preventing cast shadows. Area-wide displacement vectors were derived with the help of an image correlation technique (IMCORR, Scambos et al. (1992)): In each pair of subsequent datasets, reference patterns within a moving window of the first shaded relief (epoch n) were correlated with patterns in a defined search neighbourhood of the second shaded relief (epoch n + 1, Fig. 3a). The image correlation was applied at equally spaced nodes (i.e., central grid cells of the moving windows), providing a controlled subsampling of the raster data while enhancing the computational efficiency. Displacement vectors were then derived based on the detected positional shifts of the reference patterns. The IMCORR algorithm uses the shaded reliefs for the 2D pattern matching as well as the DSMs, so that a vertical displacement component can be added. Hence, the final output of the IM-

CORR analysis were 2.5D displacement vectors covering the active rock glacier and surrounding stable areas for each pair of subsequent DSMs. The total area covered varies for each survey campaign (Table 2).

Mean annual velocities (m/a) between the individual epochs were calculated by dividing the 2.5D displacement vector lengths by the respective time period. The resulting vectors of mean annual velocity were filtered semi-automatically to only consider downslope movement and remove minor outliers in stable areas. To allow a comparison with the results of the DGNSS-monitoring, the mean velocities derived from the image correlation analysis were spatially aggregated within the movement range of the monitored blocks (area around the block positions as shown in Fig. 1).

As an uncertainty assessment, the East-West and North-South components of the velocity vectors on stable ground were analysed individually for each period. The result of this is an uncertainty estimate for the velocity vector field over the rock glacier in East-West and North-South direction for each period. On the rock glacier surface, displacement vectors show a uniform direction due to the movement of the rock glacier (Fig. 3, box 1). In contrast, pattern shifts on stable ground outside of the rock glacier are minor and the derived velocity vectors are small and mostly show arbitrary directions (Fig. 3, box 2). Arbitrary directions of vectors on stable ground indicate random noise in the data, whereas a non-random distribution of vectors on stable ground indicates higher errors, e.g. due to shifts in the registration. The focus of this study is on the velocity of the rock glacier, hence the uncertainty analysis is centered on this aspect. Please see the references in Table 2 for more detail on absolute uncertainties of the underlying topographic data.

In order to show patterns of elevation gain and loss, differential digital surface models (DDSMs) were computed by subtracting the DSMs of two consecutive epochs (Williams, 2012). DDSM uncertainty was assessed by computing the 2.5% and 97.5% quantile of the elevation difference within stable bedrock outcrops (Table 2). This provides an estimate of the inherent noise and, hence, the detection threshold for obtaining significant surface changes (Williams, 2012). The analyses of the multi-temporal DSMs and the displacement vectors were conducted with the R statistical programming language (R Core Team, 2021).

### 2.3.3   Data fusion approach to generate time series of block rotation

The movement of destabilized rock glaciers is described as landslide-like and may have translational as well as rotational components (Buchli et al., 2018; Marcer et al., 2021). While translational movement is relatively well documented in kinematic rock glacier monitoring, few data on rotational movement are available. To assess potential rotational movement in the recently destabilized section of HEK, we analysed the rotational movement of individual blocks in the profile lines. In 2021, the DGNSS-measurements of block profiles and the ULS campaign were conducted only three days apart. This temporal proximity of the measurements made it possible to identify individual blocks from the profiles in the ortho-images generated from the 2021 UAV data (resolution: 3 cm). Unique block identifiers were manually assigned to the distinct block shapes in the respective ULS point cloud. For each of these selected point ID groups, the following analyses were carried out:

1. We applied an IMCORR image correlation at the center position of a given block in the 2021 data using the 0.1 m shaded reliefs of 2021 and 2020, in order to reconstruct the position of the block in the previous epoch (2020). Starting from the

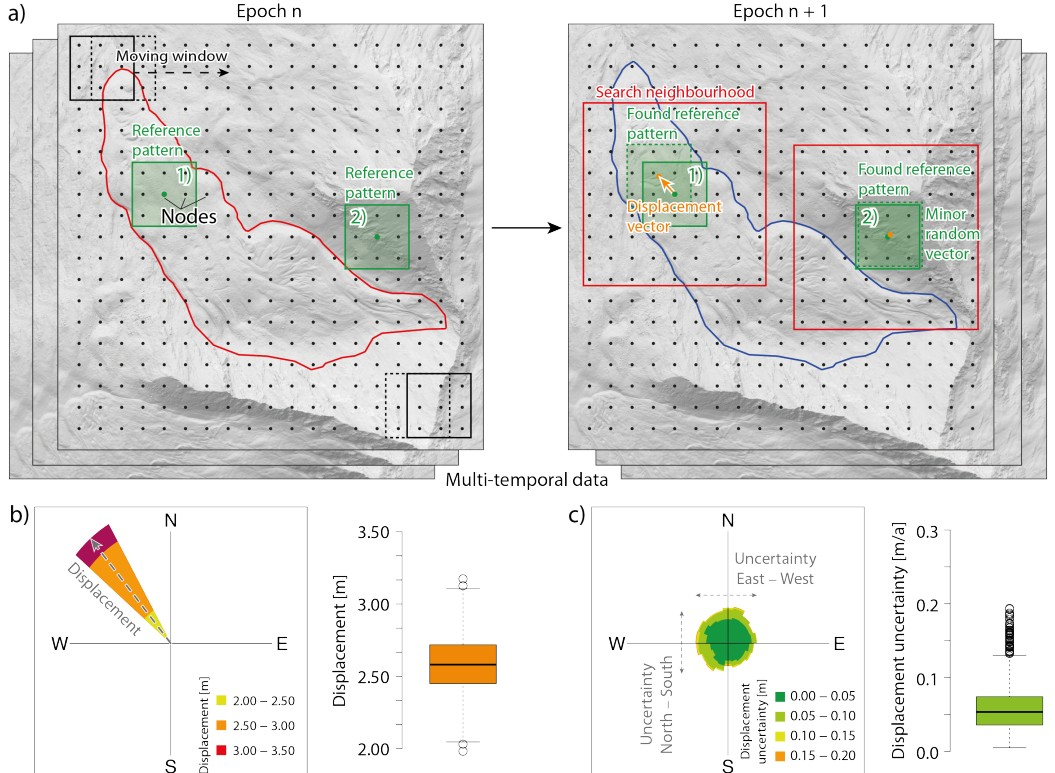

**Figure 3.** Concept of the applied image correlation approach. a) In the shaded relief of the first epoch n, a box-shaped reference pattern is analysed for all regularly distributed node positions. The green boxes 1 and 2 are the reference patterns for two example nodes, analysed during this process. For the second epoch n + 1, the reference pattern is matched (dashed box; significant shift for node 1, minor shift for node 2). b) The distribution of resulting displacement vectors' direction in epochs n and n + 1 on the rock glacier, at node 1. c) The distribution of resulting displacement vectors' direction between epochs n and n + 1 outside of the rock glacier, at node 2. The boxplots in b) and c) present an example of ranges of the derived vector lengths, showing that the uncertainty of the data on stable grounds is only minor compared to the resulting displacements.

resulting 2020 position, the same procedure was repeated to find the 2019 and 2018 positions. For each interval, this led to a 3D translation vector (x, y, z) describing the estimated block movement between two epochs.

2. Considering this translation for the initialization of a 4-by-4 transformation matrix, we used the ICP algorithm (implemented in C++ in an extension of SAGA GIS, as above) on a block-by-block basis in order to optimize the alignment of the block shapes between two consecutive epochs. This was done by a 6-parameter transformation optimising translational and rotational components. The resulting 4-by-4 transformation matrix describes the full-3D transformation of a block (assumed to be a rigid body).

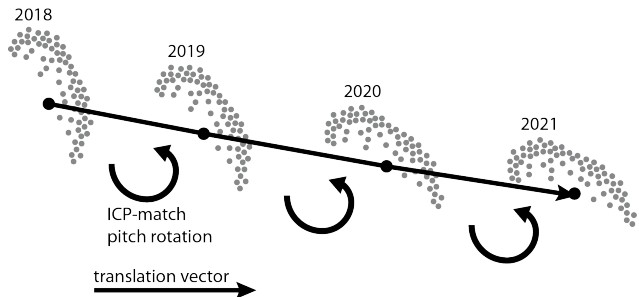

**Figure 4.** Principle of block shape matching in the ULS point clouds (grey dots) between consecutive time steps (2018-2021). The iterative closest point algorithm (ICP) was used to match the shape of a block in one epoch onto its shape in another epoch. A rotation in the opposite direction of the translation can be recognized for the block shapes. The figure shows examples of downslope translation and upslope rotation.

3. Finally, the derived matrix allowed decomposition of both the translational and rotational components of the transformation. For better interpretation, the rotational components were derived relative to the translation: the pitch rotation was defined as being in the same or opposite direction of translation, and the roll rotation was defined as being around the translation vector (Fig. 4).

## 2.4 Bulk creep factor

To interpret the observed rock glacier velocities from a dynamic perspective, we computed the Bulk Creep Factor (BCF) as described by Cicoira et al. (2020). This dimensionless parameter provides a quantitative basis for the analysis of the rheological properties of the rock glacier material, disentangling the geometrical component from the velocity signal. The calculation of the BCF is based on a modified version of Glen's flow law (Glen, 1955) adapted for rock glaciers (Arenson and Springman, 2005; Arenson et al., 2002; Cicoira et al., 2020). In the adapted flow law, the strain rates are a function of the rock glacier's slope angle, thickness, and mechanical properties. In general terms, high BCF values (roughly above 10) indicate destabilized rock glaciers with anomalous short-term deformation processes in the shear horizon dominating over long-term secondary creep in the permafrost material. However, the BCF includes both components (ice-rich core and shear horizon) and remains a proxy value for the heterogeneous properties of rock glacier material, which should be considered in detail for each case. In combination with changes in surface strain rates, spatial and temporal patterns in surface velocities, and morphological destabilization signs, discontinuities in BCF provide a further indication of the onset of a rapid sliding-type movement.

We computed the BCF for the four cross profiles shown in Fig. 1 using the time series of the respective mean profile velocities as obtained through the block displacement measurements (Section 2.1). The slope angle at each profile is given by the mean angle along the line between the fixed points that define the respective profile. The slope angle was extracted from the DSM time series data (Section 2.3.2) resampled to a 10 m x10 m grid. The resampling to a larger grid size reduces the influence of variability at the scale of surface features (such as furrows and ridges or single blocks) which are not representative of rock glacier geometry in terms of dynamics. The slope angle between DSM epochs was linearly interpolated for years in

which DGNSS block measurements are available but DSMs are not. Rock glacier thickness is given by a map of the rock glacier's bedrock extrapolated from GPR data and presented in Hartl et al. (2016a). This is a strong simplification as the depth of the bedrock may differ substantially from the depth of the thermally defined permafrost and the thickness of the moving mass. Nonetheless, lacking more detailed subsurface information on layering and potential shear horizons, we consider the approximate bedrock depth the best available estimate for our application. For parameter calibration, we used the same values as Cicoira et al. (2020), which were calibrated for a dataset of rock glaciers mostly in the French Alps. This approach allows us to directly compare our data with their results.

## 2.5 Assessment of geomorphological destabilization features

The evolution of velocity and elevation change patterns derived from the DSM and DDSM time series, along with BCF, surface strain rates and in situ velocity data can indicate destabilization onset or the end of a destabilization period (Marcer et al., 2021). Geomorphological destabilization signs - visible changes of surface morphology, such as cracks, scarps, and crevasses - are a further indicator of destabilization onset. Tracking their appearance and change over time in the DSM time series is therefore of interest, particularly when considered in combination with other potential indicators of destabilization. In the following sections, we consider morphological destabilization signs in conjunction with velocity and elevation changes for each epoch. The evolution of particular scarps in zones of the rock glacier where destabilization signs repeatedly occur was tracked and visualized throughout the DSM times series by plotting elevation profiles along the central flowline. This yields a quantitative delineation of scarp depth and downslope movement between epochs, in addition to a qualitative, visual assessment of the development of destabilization signs. The geomorphological analyses are focused particularly on the area around the terrain step at about 2570 m (zone "A") and a second zone with prominent destabilization features lower on the terminus (zone "B"), around 2520 m.

Table 2: Metadata for the presented topographic data. Values for DDSM uncertainty and level of detection refer to the DDSM or DSM pair, respectively, of the given and the previous year in m/a.

| Long-term time series of topographic change | | | | | |
|---|---|---|---|---|---|
| Acquisition date (yyyy-mm-dd) | Data source | Spatial coverage | DDSM uncertainty (i.e., 2.5 % and 97.5 % quantiles of elevation difference in stable outcrops) | Level of detection for velocities (i.e., median velocities in stable areas in x, y, z directions) | Reference |

| | | | | | |
|---|---|---|---|---|---|
| 1953-08-31 | Aerial photographs (analogue aerial stereoscopic pairs, Federal Office of Meteorology and Surveying, BEV) | Entire rock glacier | | | Klug (2011) |
| 1971-08-18 | Aerial photographs (analogue aerial stereoscopic pairs, BEV) | Entire rock glacier | -0.07 / +0.08 | -0.01 / 0.04 / -0.03 | Klug (2011) |
| 1977-09-07 | Aerial photographs (analogue aerial stereoscopic pairs, BEV) | Entire rock glacier | -0.21 / +0.17 | -0.03 / -0.16 / 0.03 | Klug (2011) |
| 1990-10-10 | Aerial photographs (analogue aerial stereoscopic pairs, BEV) | Entire rock glacier | -0.11 / +0.12 | 0.01 / 0.03 / -0.01 | Klug (2011) |
| 1997-09-11 | Aerial photographs (analogue aerial stereoscopic pairs, BEV) | Entire rock glacier | -0.18 / +0.21 | -0.01 / 0.05 / -0.01 | Klug (2011) |
| 2006-08-23 | ALS flight campaign, government of Tyrol | Entire rock glacier | -0.19 / +0.13 | 0.04 / -0.01 / 0.02 | Land Tirol Abteilung Geoinformation (2011, 2019) |
| 2009-09-09 | ALS flight campaigns carried out within the ACRP (Austrian Climate Research Programme) project C4AUSTRIA (project no:384 A963633) | Entire rock glacier | -0.13 / +0.18 | -0.01 / 0.00 / 0.00 | Bollmann et al. (2012); Klug et al. (2012) |
| 2010-10-09 | ALS flight campaigns carried out within the project MUSICALS of the Centre of climate change adaption strategies(alpS) | Entire rock glacier | -0.33 / 0.31 | 0.01 / -0.09 / 0.02 | Roncat et al. (2013a, b) |

| | | | | | |
|---|---|---|---|---|---|
| 2011-10-03 | ALS flight campaigns carried out within the ACRP (Austrian Climate Research Programme) project C4AUSTRIA (project no:384 A963633) | Entire rock glacier | -0.33 / +0.35 | -0.01 / 0.06 / -0.01 | Bollmann et al. (2012, 2015); Zahs et al. (2019) |
| 2017-09-15 | ALS flight campaign, government of Tyrol | Entire rock glacier | -0.06 / +0.06 | 0.01 / 0.00 / 0.00 | Land Tirol Abteilung Geoinformation (2011); Rieger (2019) |
| 2018-07-30 | ULS | Terminus | -0.16 / +0.28 | 0.07 / 0.15 / -0.02 | This study |
| 2019-08-30 | ULS | Lower section | -0.21 / +0.11 | 0.00 / -0.04 / -0.01 | This study |
| 2020-09-18 | ULS | Lower section | -0.09 / +0.08 | 0.00 / -0.02 / -0.01 | This study |
| 2021-08-13 | ULS | Lower section | -0.07 / +0.16 | -0.04 / -0.01 / -0.01 | This study |

**Other data**

| | | | | | |
|---|---|---|---|---|---|
| 2019, bi-weekly during the summer | TLS | Terminus | Alignment error between point clouds: 0.011 m - 0.013 m (*) | / | Zahs et al. (2022b, a) |
| 2020-09-18 | UAV ortho photos | Lower section | / | / | This study |
| 2021-08-13 | UAV ortho photos | Lower section | / | / | This study |

*Alignment error is assessed by calculating the standard deviation of M3C2 distances on stable bedrock outcrops distributed around the rock glacier (Fey and Wichmann, 2017; Zahs et al., 2022a)

## 3   Results

### 3.1   Historical timeline of terminus destabilization

The first result we focus on is the updated time series of surface displacement at HEK and the homogenisation and extension of the DSM time series for the site, which now covers a time period of 68 years. We present analyses of the multi-temporal DSMs alongside the block displacement measurements. The early data shows a destabilization phase of the rock glacier from onset to deceleration, with a peak in the 1960s (Section 3.1.1). This is followed by a period of relative stability, which includes the onset of renewed acceleration in the mid-1990s (Section 3.1.2). Detailed data on the block profiles for the early years of the

time series have been presented in previous studies (Schneider, 1999a; Schneider and Schneider, 2001; Hartl et al., 2016b). We include the mean profile velocities here again in brief to contextualise the in situ data with the multi-decadal DSM time series and morphological observations based on the DSMs. The high-frequency, high-resolution 3D data collected in the past 5 years are presented separately in Section 3.2.

#### 3.1.1   First period of destabilization: 1953 to 1977

From 1956/57 onwards, the frozen mass of the rock glacier entered a period of acceleration. Considering the block profiles (Fig. 5), P1 and P2 arguably showed irregular behaviour from the beginning of the respective time series (1955 at P1, 1952 at P2). However, this signal is hard to interpret since there is no prior data it can be compared to. Velocity vectors derived from the 1953 and 1971 DSMs reach values of 1-2 m/a in roughly the lower half of the rock glacier, up to the area between P2 and P3 (visible in the velocity vector maps in Fig. 6). It should be noted that this DSM pair does not resolve the terminus well due

to shading effects in the underlying aerial imagery. Hence, the DSM pair for this epoch likely does not capture the full range of velocities in the lowest part of the rock glacier.

In 1971-77, velocity vectors of more than 5 m/a were recorded at the terminus. In the upper part of the rock glacier, the measurements show a decrease in velocity compared to the 1953-71 period. The 1971-77 DSM pair shows that the fastest moving part of the rock glacier was below P1 during this time (Fig. 6). Comparing the velocities obtained from the DSMs with

the mean block profile velocities for this period, the velocity of profile P1 seems low compared to the maximum values of the velocity vectors. There was no block profile in the lowest section of the terminus at this time, so the DSM-derived velocity vectors show processes at the terminus that were not captured by the in situ monitoring.

The highest mean profile velocities during this first period of acceleration were recorded in 1961/62 at P1 and P2. Single blocks reached a maximum velocity of 6.6 m/a at P1 and 2.2 m/a at P2 in this measurement year (Schneider, 1999a). The next

measurement was carried out in 1970. Between these two measurements, mean profile velocities decreased from 3.9 m/a to just under 1.8 m/a for P1, and from 1.5 m/a to 1 m/a at P2 (Schneider, 1999a). P3 experienced a slight increase in velocity at the same time as the lower two profiles, as well as a slight decrease after 1969/70. However, changes were minor and, in contrast

to P1 and P2, do not clearly stand out from the later years of the time series. P1 shows relatively high BCF values during the 1960s and early 1970s (up to 12.8) and values at P2 were also elevated. At P3, BCF was only marginally higher during the period of accelerated movement than during the subsequent, more stable period (Fig. 5).

From a geomorphological perspective, the first signs of destabilization are already visible in the earliest data: In the 1953 DSM, an isolated scarp can be seen at around 2580 m.a.s.l. (close to zone "A" in Fig. 7 and Fig. 8). In the 1971 DSM, the scarp is a few tens of metres further downhill and has notably increased its size in the centre of the rock glacier body. Below this area, several other destabilization signs are visible from 2580 to 2480 m.a.s.l. The most prominent scarp (Fig. 7, zone "B") appears at about 2520 m.a.s.l., with a width of more than 150 m across and almost 300 m following the crown. The highest elevation difference is observable on the orographic left side, with more than 30 vertical metres between the crown and the top of the destabilized body. Nearby, several other cracks and scarps are clearly visible. In the following years, the destabilization signs rapidly change in size and geometry and move downslope. By 1977, the scarp in zone "A" is almost stable, while the lower area below zone "B" continues its evolution especially towards the rock glacier front. A series of profound scarps develop here in a short time span, concurrent with a notable advance of the terminus (Fig. 8, Fig. 7, Fig. S1). At the front, the oversteepened slope shows the last destabilization signs of the first phase with the onset of the lowermost scarp in 1977. In total, the front advanced roughly 115 m $\pm$ 10 m horizontally and 50 m vertically between 1953 and 1977. The 1953-71 and 1971-77 DSM pairs clearly show a pattern of elevation loss below zone "B" and concurrent elevation gain at the lowest end of the advancing terminus (Fig. 9).

### 3.1.2  Intermediate period of relative stability: 1977 to 2017

From the mid-1970s until the later half of the 1990s, the displacement rates at the surface of the rock glacier stagnated in a narrow range with low variability. The mean block profiles and the velocity vectors derived from the 1977-90 DSM pair show similar values (between 0.3 m/a and 0.7 m/a for the mean block profile velocities and between 0.2 m/a and 1.7 m/a for the 1977-90 DSM pair at the profile locations, Fig. 5). There are larger discrepancies between the mean profile velocities and the DSM derived velocities at the profile locations in the first years of the time series due to lower performance of the DSM matching algorithm and the low density of the velocity vectors in the area around the profiles (Fig. 6). The comparatively poor quality of the data results from issues with the underlying aerial imagery for this period (Section 2.3.2, Klug (2011)). During the following periods (1990-onwards), the results from the DSM matching improve in the area of the profiles (Fig. 6) and so does the correspondence with mean block profile velocities (Fig. 5). Starting in the late 1990s, the velocities at all four profiles show a clear increasing trend, accompanied by large inter-annual variations. In the 1997-2006 DSM pair, the area of elevated velocity values extends approximately from the terrain step at 2570 m.a.s.l. to P3, with maximum values of up to about 3.5 m/a recorded near P1 on the orographic right side of the rock glacier. Velocities at the terminus did not noticeably rise compared to the 1990-97 period. The 2006-09 DSM pair shows a very similar velocity pattern (Fig. 6). In the following year (2009-10), velocities increased in the area slightly below and above the terrain step up to close to P3. This trend continued in the 2010-11 DSM pair, with elevated velocities of over 1 m/a recorded above P3 on both lobes of the rock glacier. For all the profiles, two

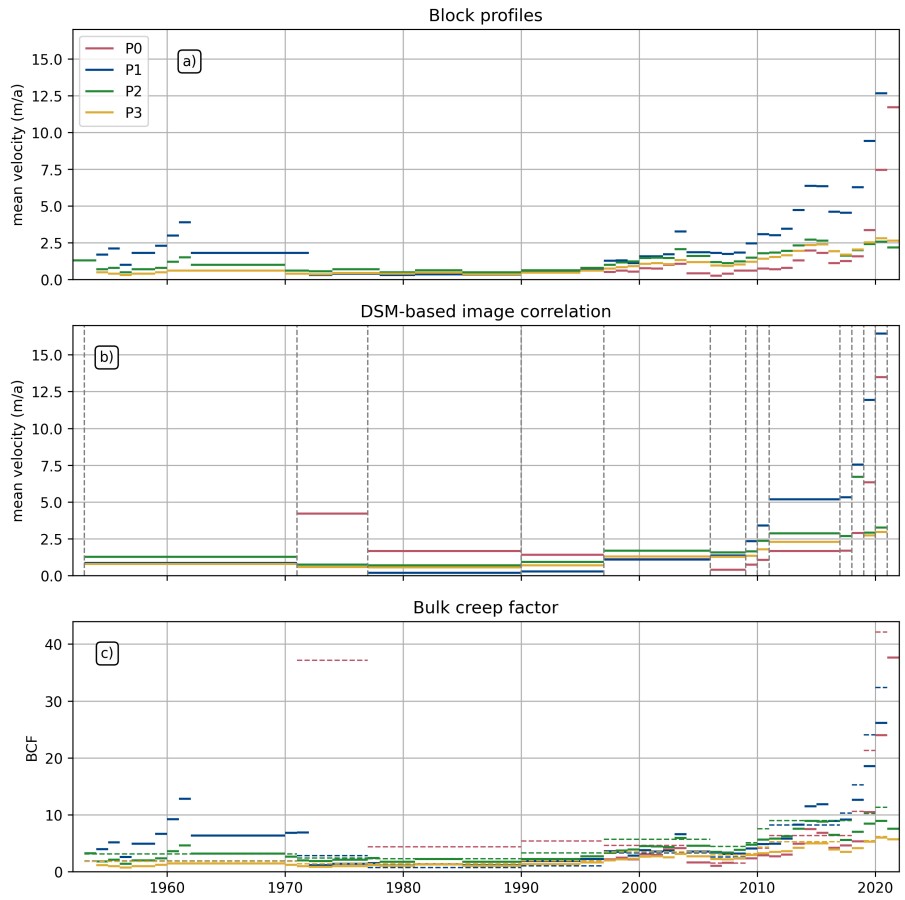

**Figure 5.** Time series of: a) mean profile velocities (m/a) at the 4 block profiles (Uncertainty estimate for the block profiles: ±0.2 - ±0.5 m/a depending on the year, see Hartl et al. (2016b)). 2022 value not shown for P1 due to loss of numerous blocks. b) Mean velocities at the locations of the stone profiles derived from image correlation of DSM pairs, vertical dashed lines indicate years for which DSMs are available. See Fig. 15 for corresponding boxplots of velocity uncertainties. c) Bulk creep factor (BCF) computed for the mean block profile velocities (solid lines) and the DSM velocities (dotted lines).

velocity peaks were reached in 2004 and in 2015-16, with 3.3 m/a and 6.4 m/a respectively at P1. A velocity minimum was reached in 2007 at P0 and 2008 at P1, P2, and P3 (0.3 m/a at P0, 1.7 m/a at P1).

The BCF remained relatively constant at P1, P2, and P3 from the mid-1970s until the mid-1990s, at what may be considered a baseline value for the respective profiles in stable conditions (Fig. 5). This stable BCF value is about 2 at P2 at about 1 at P1 and P3 based on the mean profile velocities. P0 was not established yet during this time, but computing the BCF from the DSM-derived velocities for the area of P0 indicates a higher stable BCF of 4-5 in this section of the rock glacier. The 2004 velocity peak and subsequent decrease in velocity translates to a similar pattern of BCF, which is least pronounced at P3.

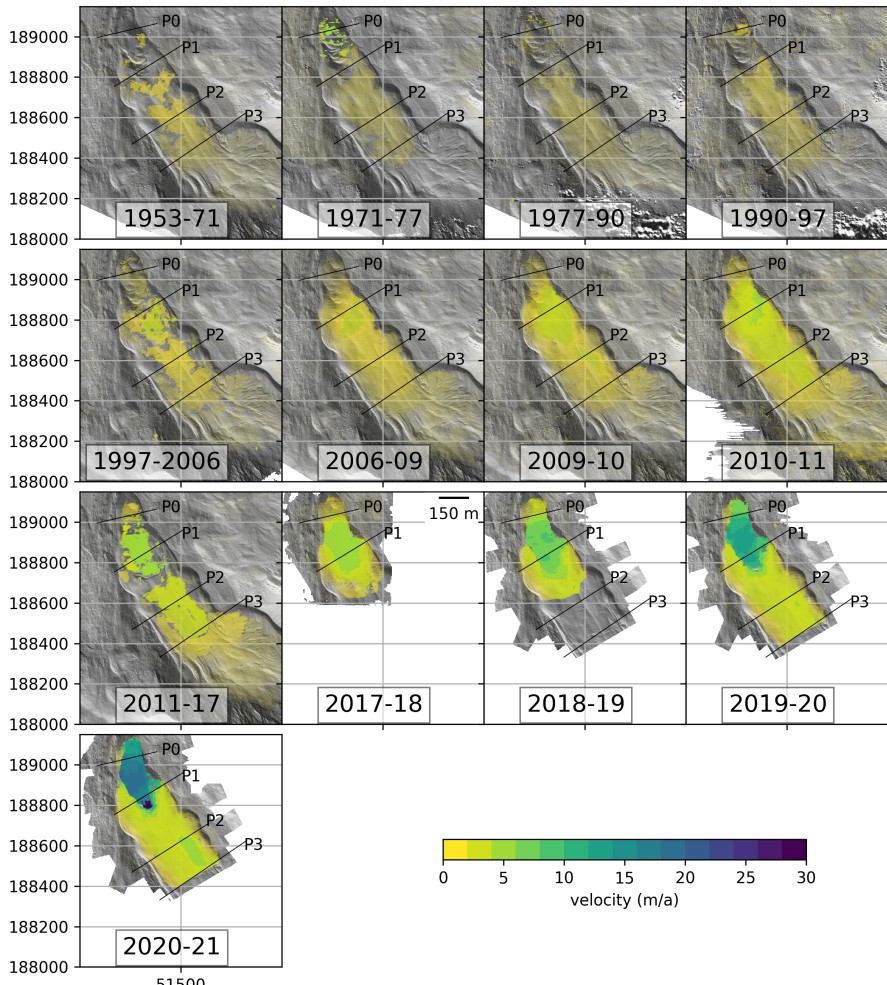

**Figure 6.** Velocity vectors (m/a) for the time series of DSM pairs, plotted over hillshades of the later DSM of each pair. Reference lines defining block profiles P0-P3 added for orientation. (Coordinate grid: EPSG: 31254)

Interestingly, BCF at P2 is very close to BCF at P1 during the 2004 peak, although velocities are substantially higher at P1. During the 2015-16 peak, BCF at P1 rose to values comparable to the first period of destabilization in the 1960s.

The morphological signs of destabilization remained mostly unchanged until the 2011 DSM, as visible in Fig. 7 and in the movie in the supplement. However, some surface evolution is visible especially in the central part of the destabilized area, where the terrain is steep and relatively large surface velocities were recorded even in this period. Between 1977 and 1990, the upper scarp shifted only minimally further downhill. The large, secondary scarp in zone "B" and the smaller scarps lower down on the terminus became less prominent over time, reducing their length and height over the years (see Fig. 8 and Fig. 10).

When analysing the elevation difference between the top of the displaced material and the low point of the secondary scarp

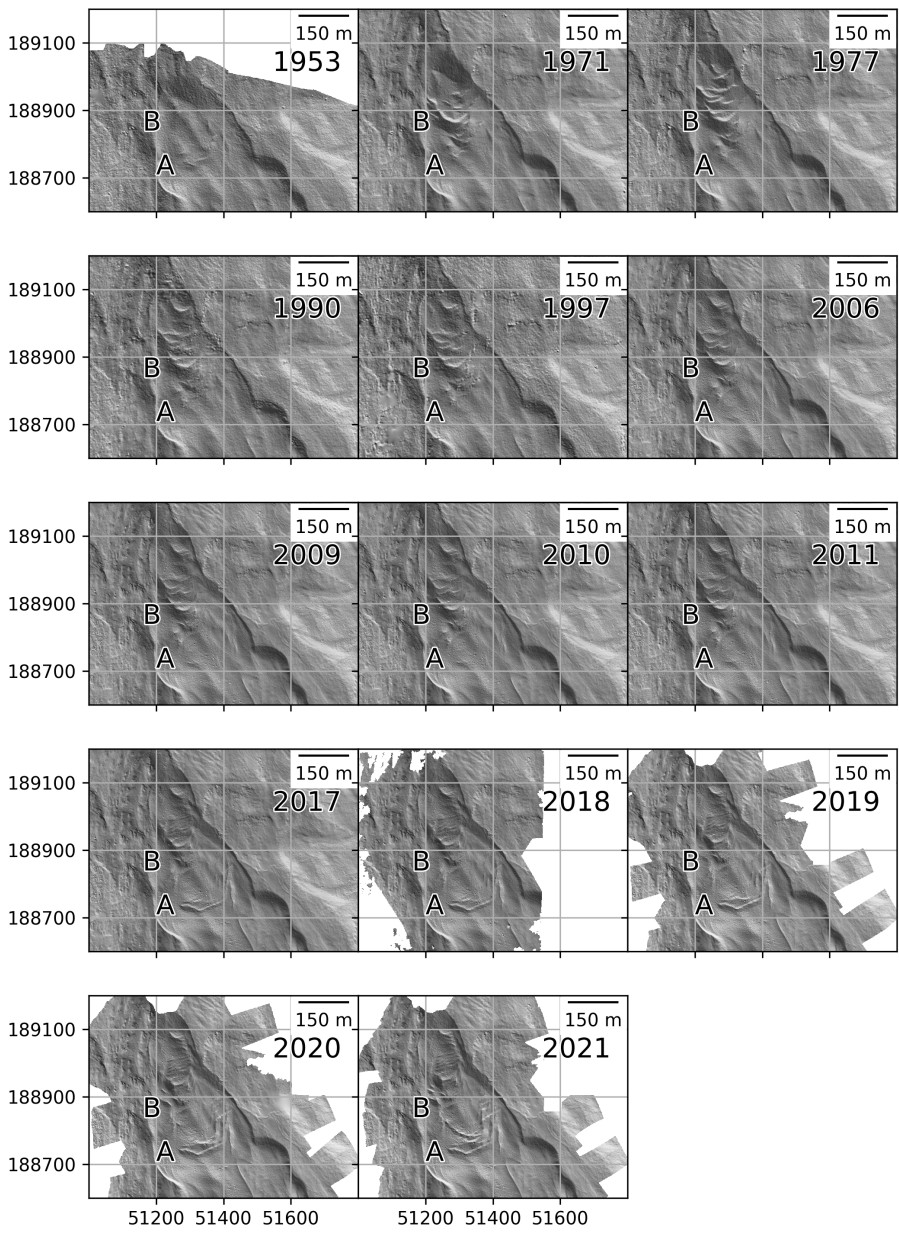

**Figure 7.** Hillshades of DSMs from 1953 to 2021. A and B mark upper and lower sections of morphological destabilization signs as referred to in the text. An animated version of the time series of hillshades is available in the video supplement. (Coordinate grid: EPSG: 31254)

(Fig. 10), it is evident that the smoothing process developed slowly up to the 1997 DSM. Scarp height decreased by about 3 m in almost 40 years (Fig 10). Between 2011 and 2017, scarp height decreased by about another 2 m as velocities increased in

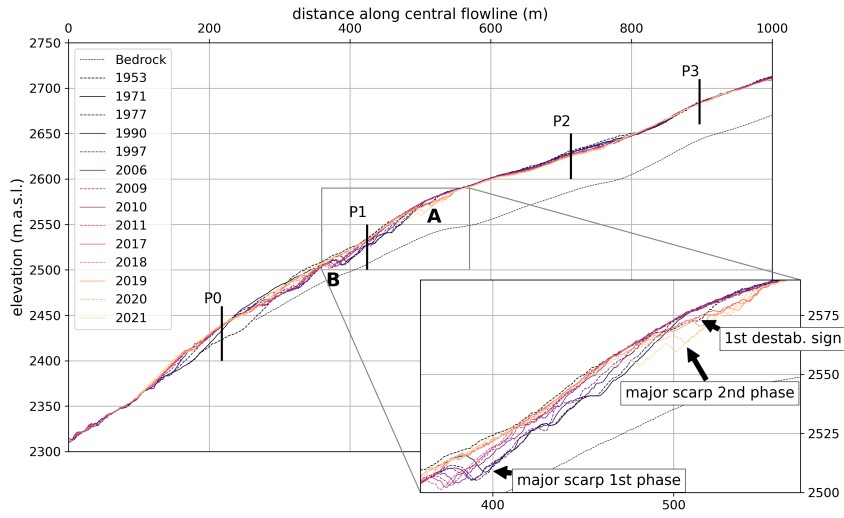

**Figure 8.** Surface elevation along the flow line as extracted from the DSMs. Estimated bedrock profile (dashed line) from Hartl et al. 2016b. Annotations show positions of the cross profiles. A and B mark upper and lower zones of morphological destabilization signs, as in Fig. 7.

the area above the scarp and the scarp moved downhill, shifting over 10 m horizontally (Fig 10, b). Similarly, the majority of the other destabilization signs on the terminus gradually smoothened (Fig. 8). The advance of the terminus in this period was

small compared to the previous, more unstable period. The 1977-90 and 1990-97 DSM pairs show moderate elevation gain at the very end of the terminus and elevation loss directly above this area (Fig. 9). Later, in 1997, 2006, 2009, 2010, and 2011, an area of positive elevation change can be seen between about the elevation of P1 and just above the terrain step. The upper part of the rock glacier mostly shows slightly negative elevation change, with the magnitude of negative change increasing in the later DSM pairs. The signal of elevation change in the terminus area is characterized by small scale variations around

individual morphological destabilization features shifting downhill (Fig. 9). The front of the smaller, less active orographic right lobe shows positive elevation change due to a gradual advance.

### 3.2   Recent destabilization observed from high-resolution monitoring

In this section we take a closer look at the data since 2017, for which a much higher temporal and spatial resolution is available. We extend the previous analysis and additionally focus on the movement and rotation of single blocks derived from 3D point

clouds, as well as sub-seasonal changes at the rock glacier front. High-resolution 3D point clouds were derived from state-of-the-art close-range sensing techniques (ALS, ULS, TLS, see Table 2). These datasets are spatially very highly resolved with a spacing of the individual point measurements in the order of centimetres. Hence, they allow a detailed study of the kinematics and geomorphological evolution of the rock glacier. Since 2017, surface change at the rock glacier is characterized by unusually large displacement rates and by the development of a second destabilization phase in the lower part.

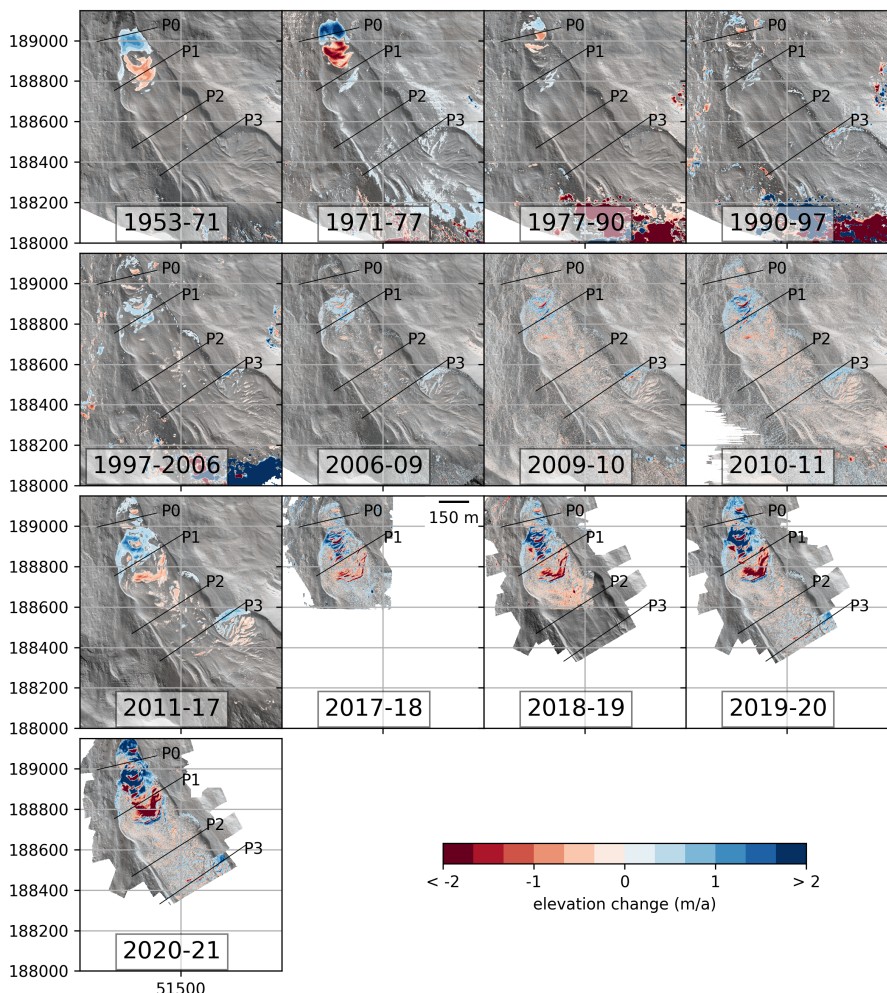

**Figure 9.** Vertical surface elevation change (m/a) beyond the individual uncertainty (Table 2) for the time series of DSM pairs, plotted over hillshades of the latter DSM of each pair. High change values in the steep rock face above the root zone of the rock glacier are artefacts from the photogrammetric processing due to shadows and/or snow cover in the images and are included for the sake of transparency. Reference lines defining block profiles P0-P3 added for orientation. (Coordinate grid: EPSG: 31254)

### 3.2.1 Kinematics

In measurement year 2018/19, velocities at the block profiles started increasing again after the previous, short-term slow down. P1 was consistently the fastest profile throughout the time series and remained so with a mean profile velocity of 12.6 m/a in 2020/21. As of 2020/21, P0 was second fastest with 7.4 m/a. The high velocity at P0 is particularly noteworthy compared to previous years: Since its establishment in 1997, P0 was the slowest of the 4 profiles, suggesting advanced degradation of the rock glacier in the terminus area. P3, the next slowest profile, was typically faster than P0 by 0.2 m/a or more. However,

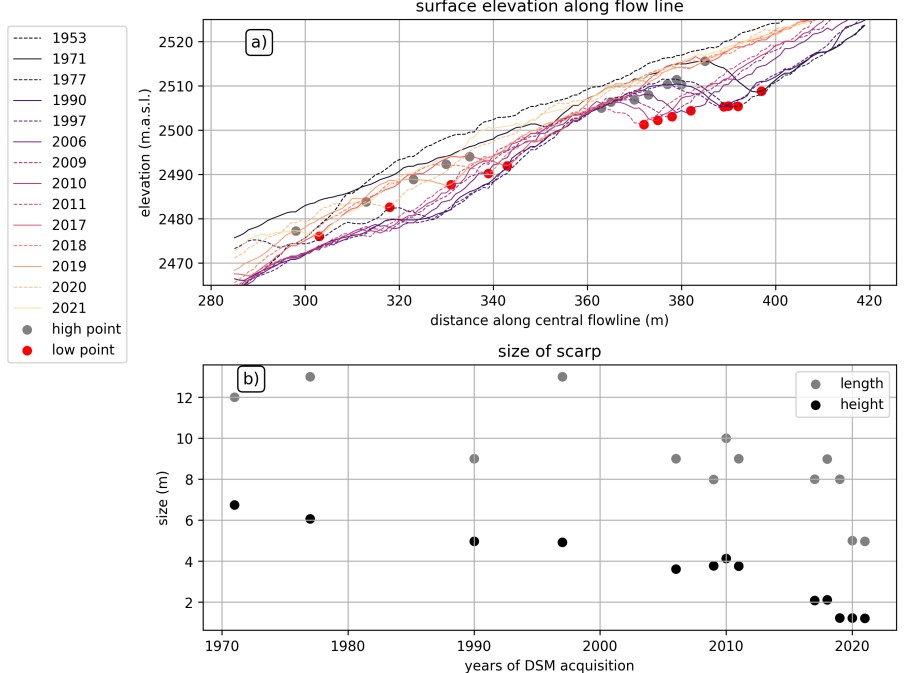

**Figure 10.** a) Surface elevation along the flow line as extracted from the DSMs, zoomed in on the scarp in zone "B", as marked in Fig. 7, and approximate locations of the bottom of the scarp and the high point of the displaced material. The scarp is not yet present in 1953. b) Height and length of the scarp in different years (height is the vertical distance between the respective grey and red dots in a); length is the horizontal distance).

in the 2019/20, 2020/21, and 2021/22 measurement years, P0 was faster than P2 and P3, deviating strongly from the pattern of the previous 2 decades (Fig. 5). BCF values reflect the shift in behaviour at P0 and jump to over 20 in 2020/21 and over 30 in 2021/22. P1 shows a similarly sharp increase in BCF. The mean profile velocity of P1 in 2021/22 is not shown in Fig. 5 because multiple blocks were lost in this measurement year and the mean of the remaining blocks is no longer considered

representative (see Fig. 11 for single block velocities). At P2 and P3, BCF has also been elevated in recent years but remains in the same range as during the 2015-16 peak. P2 and P3 slowed down slightly in 2021/22 compared to 2020/21. At P1 and P0, the 2020/21 profile mean velocities represent 567% and 582% of the time series mean, respectively. At P2 and P3 the anomaly is pronounced but less extreme (224% and 336%, respectively).

Considering the movement of single blocks in the profiles (Fig. 11), it is apparent that the large increase in mean profile

velocities at P0 and P1 is driven by blocks in the central and orographic right section of the profiles. In 2020/21, the maximum block velocities of P0 and P1, respectively, were 13.6 m/a and 20.6 m/a. In 2021/22, the maximum block velocity of P0 increased yet again to 18.5 m/a. At P1, the fastest block of 2020/21 was not found in 2021/22. Nonetheless, the maximum velocity of the adjacent block was 23.6 m/a - a new absolute maximum value for all profiles. Blocks on the orographic left and at the margins of the profiles show a far more gradual acceleration without the extreme jump in velocities in the most recent

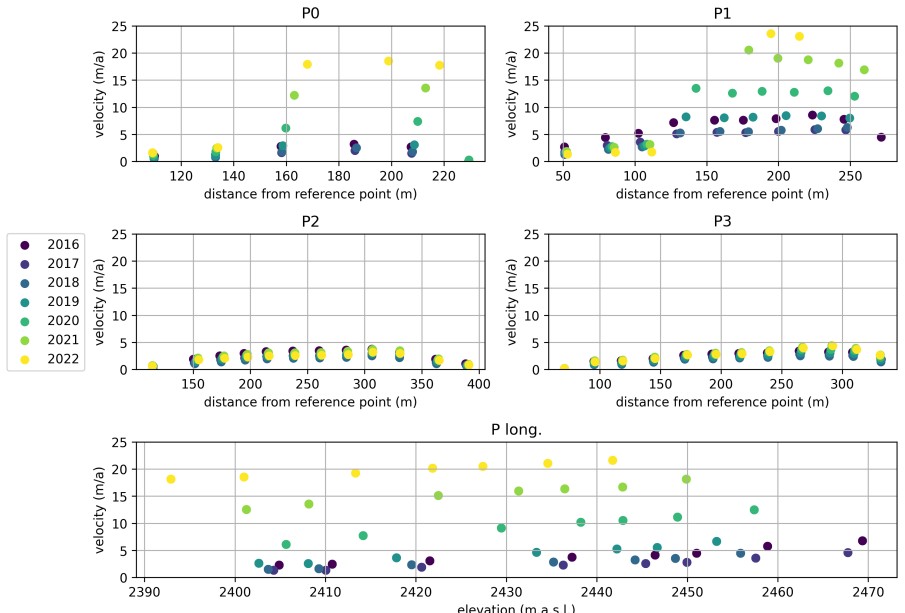

**Figure 11.** Block velocities for the four cross profiles (P0-P3) and for the longitudinal profile (P long.) in m/a for 2015/16 - 2021/22. The x-axis shows the distance from the orographic left reference point for P0, 1, 2, and 3, respectively. For P long., the x-axis shows the elevation of the blocks in each year. In P0, the third block from the right could not be located in 2018-2021, presumably because it rotated so that the marking was obscured. In P1, a block was similarly lost in 2021, and 3 more were lost in 2022. For P. long., only blocks that were found every year since 2016 are shown.

years. At P1, the far left blocks have slowed down slightly and gradually shifted further into the rock glacier area in recent years.

The highest annual 2.5D velocity detected in the DSM time series at a derived IMCORR-vector is 32.1 m/a in 2020/21 just below the terrain step (Fig. 6). This could vary slightly if the correlation analysis was performed at different nodes (see section 2.3.2, Fig. 3), so some caution is required when interpreting vector magnitudes. The velocity vectors in the DSM pairs of 2011, 2017, 2018, 2019, 2020, and 2021 show a similar pattern of velocity distribution as in the previous years, but with significantly and progressively higher velocities each year in the area at and below the terrain step. This is a strong indication that this section is central to the renewed destabilization process.

### 3.2.2 Destabilization signs

In the 2017 DSM, a large new crevasse is visible shortly upstream of the terrain step where the first destabilization phase started (zone "A"). A large section of the slope below this area was displaced by about 10 vertical metres between 2011 and 2017 (Fig. 7, Fig. 8, Section 3.1.2). Between 2017 and 2021, the crevasse developed into a scarp concomitant with the appearance of new destabilization signs in its vicinity (cracks and crevasses). Some of the cracks closer to the terrain step were initially

confined towards the left and right margins of the rock glacier. They were then connected by a long crevasse in 2019. This feature rapidly developed towards a scarp in the following years. By 2021, another large and very convex scarp appeared just a few metres downslope. The development of these destabilization signs coincided with the large velocities illustrated in the previous section. While the scarps in zone "A" widened and deepened, the large scarp in zone "B" (Fig. 10) that developed during the first destabilization cycle continued decreasing in size and moved downslope, as described in the previous section. Recent TLS data from the lowest section of the terminus also show continuous smoothing of individual morphological features in recent years (Fig. S5).

The distribution of elevation change (Fig. 9) shows a similar pattern. Starting in 2011, pronounced surface elevation gain in the lower part and elevation loss in the upper part are visible. The signal of positive elevation change rapidly (within 4 years) propagates towards the front (Fig. 8, Fig. 9).

Zone "A" lies between P1 and P2 and the surface strain rates between the two profiles also reflect the rapid changes in this part of the rock glacier in recent years. Strain rates were in a relatively low range of up to about 0.005 $a^{-1}$ until 2011. Strain rates then increased at blocks in the central and orographic right part of the rock glacier, with an initial jump between 2013 and 2014 at most of the affected blocks (Fig. 12). Until 2019, strain rates in this section ranged from roughly 0.01 to 0.02 $a^{-1}$. Then, a large jump occurred and strain rates reached values around 0.03 $a^{-1}$ in 2020 and between 0.04 and 0.05 $a^{-1}$ in 2021. In 2022, multiple blocks in P1 were lost. At the remaning blocks, strain rates increased again to near 0.06 $a^{-1}$. Strain rates remained low on the left margin of the rock glacier throughout and even decreased there slightly in the most recent years (Fig. 12).

### 3.2.3   Rotation of blocks

Individual blocks in P0, P1, P2 and in the longitudinal profile were identified in high-resolution UAV orthophotos and tracked in the associated ULS point clouds for 2018-2021, yielding rotation angles of the blocks along the axis of the flow direction. Rotation angles are generally low (roughly $\pm 1°$) at P2 and higher at P1, P0, and the longitudinal profile (Fig. 13). The blocks on the orographic left side of P1 show little rotational movement or tend to tilt forwards, while the blocks further to the right in the faster moving sector tend to tilt backwards in 2018/19 and 2019/20. In 2020/21, the rotational movement at P1 appears reduced overall. Blocks in P0 and in the lower part of the longitudinal profile where it intersects P0 tend to tilt backwards in all three years, with highest rotation angles (>5°) in the central part of the terminus near a morphological destabilization feature (Fig. 13). From 2018/19-2020/21, patterns of positive and negative surface elevation change around this feature became more pronounced. The strongly positive rotation angles are predominantly found in an area above the feature with positive surface elevation change.

### 3.2.4   Sub-seasonal displacement (2019)

For the bi-weekly TLS time series of summer 2019 we quantified 3D topographic change between two epochs by calculating CD-PB M3C2 distances between corresponding planar boulder faces (plane pairs). We considered CD-PB M3C2-based surface change in flow direction, vertical direction and horizontal direction. We quantified significant change in flow direction for 58,074 to 62,138 plane pairs, in vertical direction for 38,779 to 40,543 plane pairs, and in horizontal direction for 27,227 to

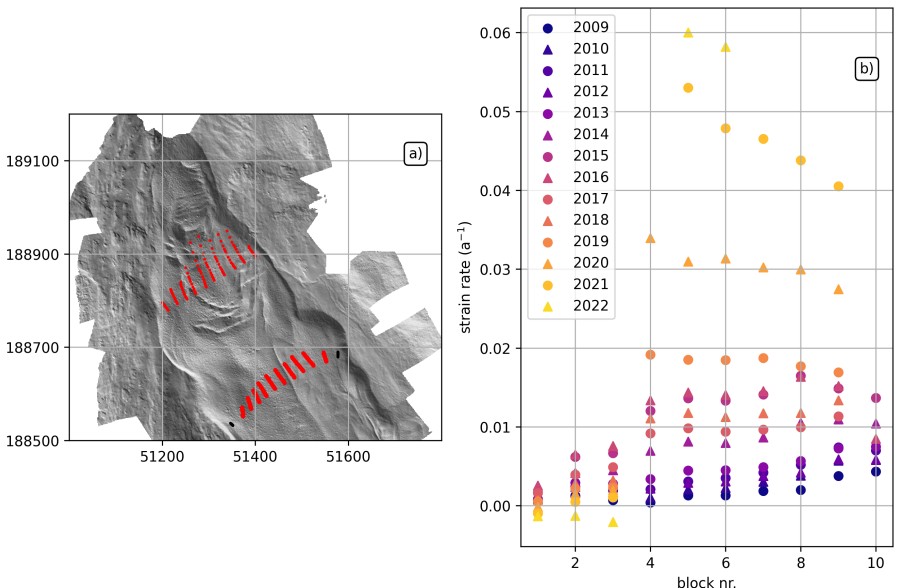

**Figure 12.** Strain rates for the blocks in P1 and P2 (excluding the furthest left and furthest right blocks in P2) for the measurement years 2009/10 to 2021/22. Panel a) shows the location of the blocks in each year (red dots) over the 2021 hillshade. Panel b) shows the strain rate per block pair, counting the blocks in the profile lines from west to east.

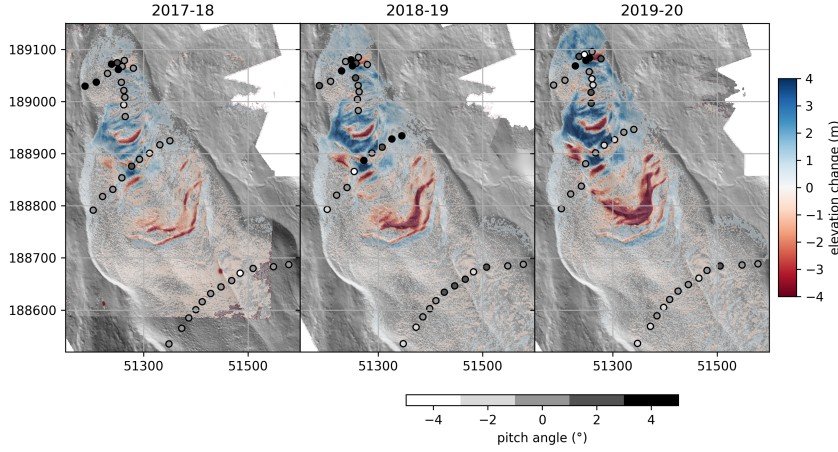

**Figure 13.** Pitch angle of each DGNSS block identified in the ULS data for the years 2018/19, 2019/20, and 20/21, plotted over the elevation change layer of the same time period and a hillshade of the later DSM. Black lines show the displacement vectors for the cross profiles during the 3 time periods.

30,078 plane pairs. In this study, the uncertainty associated with quantified change ranges between 0.014 m and 0.015 m. This allows significant surface change to be quantified in all three directions in 73.95% to 75.28% of the total area for which change quantification is applied (i.e. where corresponding boulder faces are identified). In Figures 14, S3, and S4 (supplementary material) only significant surface change is shown.

Our analysis reveals short-term variations in the magnitude of the movement along the flow direction at the rock glacier front from 0.12 m in 13 days to 0.21 m in 15 days in July and August 2019 (Fig. 14). These values correspond to average movement of 0.009 m/day in early July and August and 0.014 m/day during the second half of July and August, respectively. Standard deviation of the magnitudes is relatively constant at 0.07-0.08 m for each time step (supplementary Figures S3, S4). The magnitude of surface elevation change per time period also varies, with the largest values recorded in the first measurement

period between the last week of June and the first week of July. Vertical change between plane pairs at the rock glacier front is predominantly negative (Fig. S3).

## 4    Discussion

### 4.1    Kinematic data and change detection: Uncertainties and challenges

We follow the method of previous studies for the calculation of mean profile velocities from the locations of the marked blocks,

i.e. the profile mean is the mean of the available blocks per profile (Schneider and Schneider, 2001; Hartl et al., 2016b). This ensures the consistency of the time series. However, we acknowledge that it does not account in detail for missing data from blocks that can no longer be found, the periodic repainting and repositioning of single blocks or profile lines, missing years, or the slight year-to-year differences in measurement dates. In 2021/22, multiple blocks in P1 were lost, so that the mean profile velocity can no longer be considered representative. A new block line has been established for future use, but this highlights the

challenges of in situ monitoring during a destabilization phase and rapid changes at the rock glacier surface. The magnitudes of the velocity vectors derived from the DSM pairs and the general distribution pattern of increase and decrease in velocities agree well with previous studies, with minor discrepancies due to different node placements (Fig. 3) and methodological adjustments (Klug, 2011; Klug et al., 2012).

    The uncertainties inherent to the 2.5D velocity vectors as derived from change detection between DSM pairs are in a similar

range as was found by comparable studies at other rock glacier sites (e.g., Bodin et al. (2018); Fleischer et al. (2021); Kummert et al. (2021)). The uncertainty analysis of the velocity vectors is based on an assessment of velocity vectors within stable areas around the rock glacier, which we use as a measure of noise and systematic errors in the data. Arbitrary directions and small magnitudes of the velocity vectors over stable ground indicate small uncertainties due to random noise, while directional bias and large magnitudes indicate larger errors. We individually consider the directions of the vectors within the stable areas to

analyse directional biases of the data as well as noise and to assess data and registration quality (see also Table 2, Fig. 15).

    In East-West direction, the median velocity within the stable areas shows only minor deviations (e.g. +0.07 m/a in 2017-2018, Fig. 15a). In North-South direction however, distinct deviations emerge for the periods 1971-1977 (-0.16 m/a), 2009-2010 (-0.09 m/a) and 2017-2018 (+0.15 m/a) (Fig. 15b). Due to the topography in the area of interest, mainly East- and West-facing

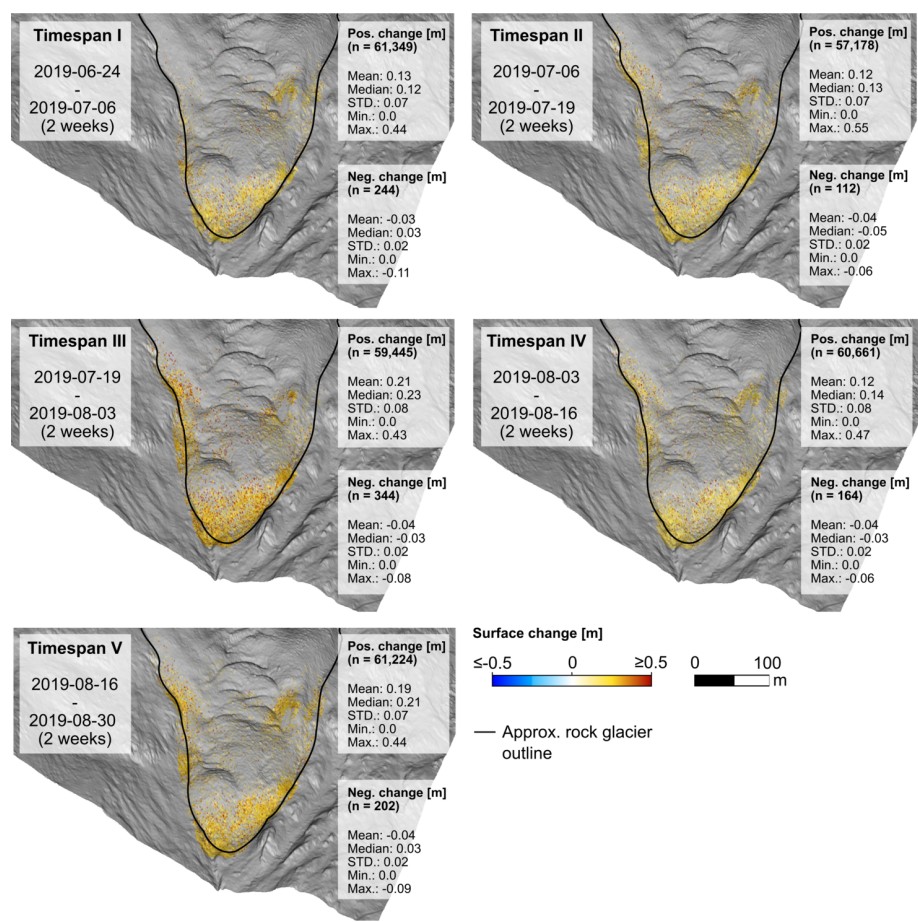

**Figure 14.** Magnitudes of surface change in flow direction derived from the correspondence-driven plane-based M3C2 for 2-week timespans. A hillshade derived from airborne laser scanning data is used as baselayer in all subfigures. As the CD-PB M3C2 algorithm favors confident detection of small-magnitude changes over full point-wise change quantification, change information is more dense in at the steep rock glacier front where high point density, spatial coverage and spatial overlap between point clouds of two epochs could be achieved with TLS-based data acquisition.

slopes are covered by the data with sufficient quality, allowing for a more robust registration in the East-West direction. North- and South-facing slopes are much less common in the East-West oriented cirque of HEK, leading to a less robust registration in North-South direction and higher uncertainties respectively. The deviations of median velocity within stable areas in the vertical direction are generally negligible, with the highest deviation in the period 1954-1971 (-0.03 m/a).

Within the scope of this study, our main interest is an assessment of the kinematics of HEK and the velocity vectors derived for each period in the DSM time series. Since periods between subsequent epochs are irregular, we normalise the displacement vectors by the time periods to obtain mean annual velocities that are directly comparable. We focus our uncertainty assessment

on these values. Please see the publications cited in Table 2 for more details on absolute uncertainties of the various datasets. We use DSMs derived from different kinds of underlying data (Fig. 2). The uncertainties of the velocity vectors as shown in Fig. 15 reflect some of this variability: DSM pairs photogrammetrically derived from scanned analogue historical aerial imagery show the highest uncertainties in terms of absolute displacement (please see Klug (2011); Klug et al. (2012) for

further detail). However, when considering velocity rather than total displacement, uncertainties are relatively low due to the longer periods between the acquisition campaigns (e.g. 1953-1971, 1977-1990). The photogrammetric DSM pairs covering shorter periods (1971-1977, 1990-1997) accordingly show higher velocity uncertainties (Fig. 15). The DSM pairs marking the transitions between acquisition methods (1997-2006: photogrammetry-ALS; 2017-2018: ALS-ULS) are also associated with higher velocity uncertainty (Fig. 15). This may be a result of the differing level of topographic detail captured by different

acquisition techniques. For example, DSMs produced using photogrammetric techniques based on historical aerial imagery typically contain less detail, and the spatial resolution increases drastically from photogrammetry to ALS and ULS. On the other hand, the higher spatial resolution of the ALS and ULS data includes more topographic detail, which will affect the aggregation of DSMs with the 1m cell size, chosen as a compromise between the historical and recent data. In the case of the DSM pair 2017-2018, the reduced spatial coverage of the 2018 ULS dataset limits the possible selection of stable areas for the

co-registration with the 2017 ALS dataset. This particularly affects the uncertainty in East-West and North-South directions (Fig. 15). Despite these caveats, we consider the quality of the time series of velocity vector fields over the rock glacier area adequate given the data basis and acquisition/processing techniques. Uncertainties are well below the derived velocities and the time series of velocity vectors provides an overview of shifting patterns of acceleration and deceleration that adds valuable spatial context to the point-scale, in situ displacement data.

The TLS dataset of the rock glacier front allows the quantification of short-term (bi-weekly) variations of 3D surface change. As magnitudes of rock glacier surface change tend to be small (< 0.1 m) at such monitoring intervals, sophisticated methods for 3D change detection are required. Such methods need to be capable of quantifying 3D surface change with low uncertainties. We were able to confidently quantify 3D surface change in different directions (flow direction, vertical direction, horizontal direction) and to reveal related sub-seasonal variations for a large number of corresponding planar boulder faces in two epochs.

The sub-seasonal dataset presented in this study mainly covers the rock glacier front, thus limiting the comparability of this data with the long-term DSM time series. Future high-frequency monitoring set-ups might integrate point clouds derived from TLS as well as UAV-borne 3D sensing techniques (UAV-borne laser scanning, UAV-borne photogrammetry) as the latter offers increased coverage and a more uniform point distribution compared to ground-based sensing techniques (Zahs et al., 2022a). This would allow detailed study of sub-seasonal variations of 3D surface change for larger parts of the rock glacier.

Combining the in situ DGNSS data of the block profiles with the remote sensing data leverages the advantages of both methods and partially compensates for their disadvantages: On the one hand, the analysis of the DSM pairs during the first acceleration period shows high velocities in the terminus area that were not recorded in the time series of the blocks, since there was no block profile at the terminus during this time. The velocity patterns in the DSM analysis provide further spatial context to the point measurements of the blocks, yielding a very high-resolution overview of the spatial progression of the

destabilization process. On the other hand, the higher temporal resolution of the block time series resolves shorter-term vari-

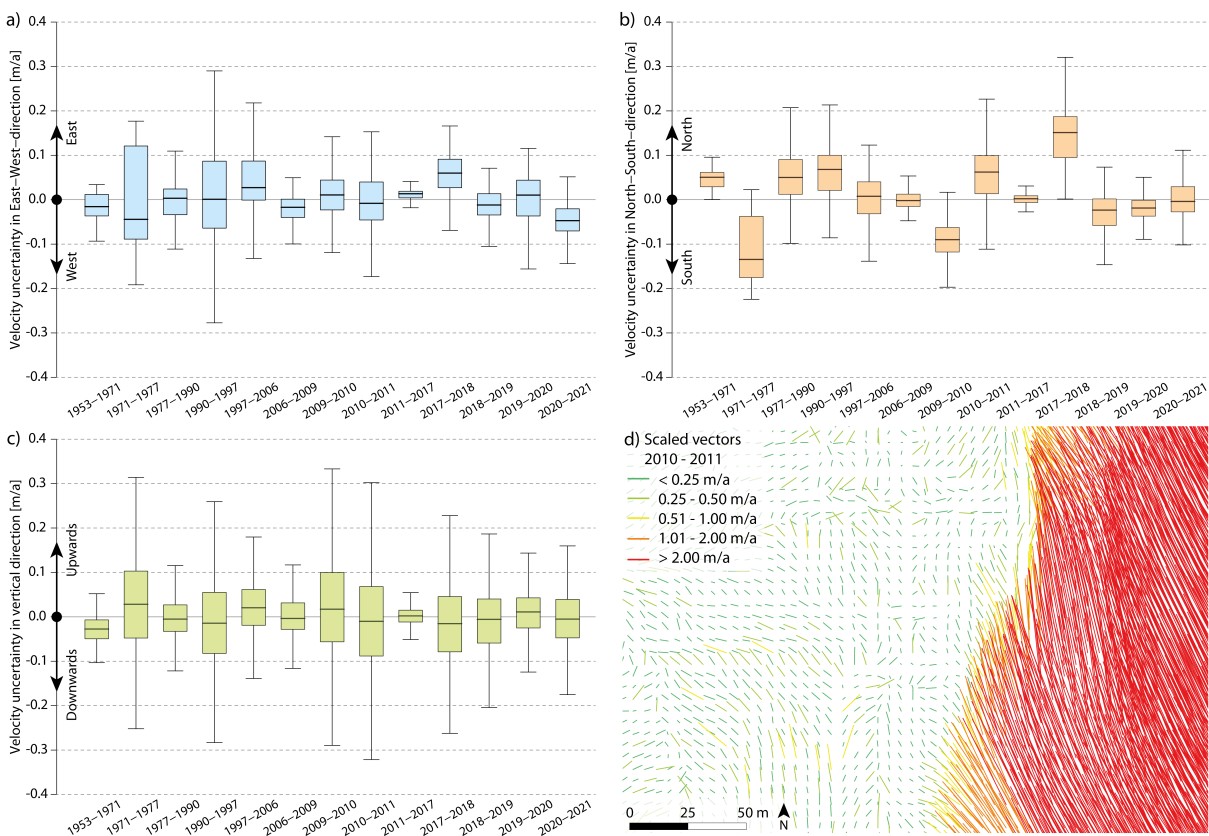

**Figure 15.** Uncertainty of velocity vectors based ob analysis of vectors on stable ground surrounding the rock glacier for each DSM-pair in East-West direction (a), North-South direction (b) and vertically (c). The boxplots show the median, the interquartile range (box) and up to 1.5 times the interquartile range (whiskers). Outliers are not shown. An example of the resulting velocity vectors (20-fold exaggerated for visualisation purposes) for the period 2010 to 2011 within and outside of the rock glacier area is shown in (d). Derived vectors on the rock glacier (yellow, orange and red colours) agree well in terms of their direction while on surrounding stable ground vectors of minor length and random direction (green colours) can be observed.

ability - notably the 2004 and 2015-16 velocity peaks - which is not apparent in the DSM data. The fusion of DGNSS data and high-resolution 3D point clouds further allows the extraction of rotational information for individual blocks, aiding the interpretation of the rotational movement of the destabilized rock glacier terminus (see following section) and highlighting the potential of multi-method monitoring. In the future, such analysis might be extended by integrating change information obtained between corresponding planar boulder faces with the CD-PB M3C2 algorithm.


## 4.2 Interpretation of results in a destabilization context

In the following, we discuss the long time series of surface displacement at HEK from a geomorphological perspective. The temporal resolution achieved with the 14 DSMs, complemented by surface velocities from block profiles from 43 measurement

years allows for a comprehensive assessment of the two phases of destabilization (see Fig. 2, first DSM from 1953; annual DSMs 2017-2021; first block data from P1 in 1952 and from P2 and 3 in 1955; annual data from 4 profiles since 1997 with a one year gap in 2005).

In the literature, the evolution of a rock glacier with regard to the destabilization process is dynamically and geomorphologically divided into four phases: normal activity (phase 1), destabilization onset (phase 2), destabilization peak (phase 3), and deceleration/inactivation (phase 4, Cicoira et al. (2020); Marcer et al. (2021)). We follow this approach and divide the two destabilization cycles observed in our dataset on the basis of a combination of displacement and surface strain rates, geomorphological signs, and the values of the BCF for different areas of the rock glacier. The earliest data in our time series coincide with the onset of the first destabilization cycle, with clear geomorphological destabilization signs rapidly developing. Despite the relatively low temporal resolution of the first part of the DSM dataset (1953-2006), the time series represents an unprecedented result for this period with three DSMs and 14 years of block velocity measurements documenting the first destabilization cycle between the early 1950s and late 1970s. As a result, we are able to distinctly map and follow the geomorphic signs as they evolve downslope and are able to resolve the acceleration and destabilization phase in the kinematic signal.

The two distinct cycles of destabilization of HEK are in general accordance with other publications that have analysed landform evolution over multi-decadal time series (e.g., Kääb et al. (2021); Marcer et al. (2021)). From these studies, the first cycle took place around the middle of the last century and the second cycle is reported approximately after the 1990s, with locally variable timing of the destabilization onset. For HEK, the onset of a second cycle of destabilization can first be definitively observed in the 2017 DSM, after a longer but more gradual acceleration phase. Increases in velocity and strain rates are apparent in the data from in situ block monitoring two to three years prior to 2017, which could suggest earlier destabilization onset. The timing of the onset cannot be determined more exactly due to the gap in DSMs between 2011 and 2017. The first destabilization cycle at the HEK started in 1953 and lasted until the late 1970s, affecting mostly the area of the rock glacier below the terrain step and leaving relatively undisturbed the terrain above. This observation is in line with the previous assessments of Haeberli and Patzelt (1982) and Schneider and Schneider (2001), who described the same general process and timeline in terms of a separation of the tongue below the terrain step from the upper, "healthy" part of the rock glacier. The surface elevation change along the flow line between 1953 and 1971 and 1977 as extracted from the respective DSMs (Fig. 9) also agrees well with the data on terminus advance and elevation change presented and discussed in Schneider and Schneider (2001).

During the first few decades of our time series, the integration of data from DSMs and block measurements shows destabilization phases 2 (onset) to 3 (peak) and 4 (deceleration, return to normal conditions). Phase 3 is characterized by high velocities and values of the BCF close to 40 in the lowest section of the rock glacier. During the subsequent period of relative stability and lower velocity values, the BCF values at P1 and P3 remain close to 1, which can be considered these profiles' "stable BCF" value corresponding to normal conditions rather than the extraordinary (Schneider and Schneider, 2001) conditions of destabilization. The renewed onset of acceleration in the 1990s did not coincide with the appearance of new morphological destabilization signs, nor with a clear and drastic dynamic decoupling between the two upper and lower sections of the rock glacier, as indicated by the low values of the BCF in both regions. After a long period of relatively continuous acceleration,

clear signs of destabilization onset (phase 2) appeared in the 2017 DSM. In terms of morphological destabilization signs, this is a clear indication that a second destabilization cycle has begun. As noted above, the initial onset may have occurred prior to 2017 based on strain rates between blocks. This cannot be determined more exactly based on morphological destabilization signs since no DSMs are available between 2011 and 2017. In the following years, velocities and in BCF at the lower profiles increased drastically. At P0 in particular, the sharp rise in velocity and BCF since 2018 suggests a fundamental shift in underlying processes. The lowest section of the terminus appeared strongly degraded and showed comparatively little activity even during the 2004 velocity peak. The recent increase in velocities at P0 is likely due to the accelerating and increasingly destabilized upper section of the terminus pushing this lower section in the flow direction. In general, the velocities of the destabilized section seem entirely decoupled from the velocity patterns in the upper part and on the margins of the rock glacier and are significantly higher than at other rock glaciers in the region (e.g., Fleischer et al. (2021)). The increasing surface strain rates across the terrain step (between blocks in P1 and P2) and in particular the very high values in the most recent years are yet another indicator of highly destabilized conditions in the lower sector of the rock glacier compared to the relatively stable upper part. The magnitudes of the strain rates are in close agreement with findings by Marcer et al. (2021), who calculated surface strain rates for destabilized and non-destabilized rock glaciers. They found that strain rates are about an order of magnitude larger in the destabilized cases. The large jumps in strain rates at HEK in 2020 and 2021 across zone "A" are likely caused by the formation and evolution of the newly developed scarps in this area. In a study assessing the surface movement of Gruben rock glacier (Switzerland) Haeberli et al. (1979) suggested that the critical strain rate for the formation of crevasses is lower for frozen debris material ($2.7 \pm 0.9 \; 10^{-3} \; a^{-1}$) than for massive ice ($1.4 \pm 1.0 \; 10^{-2} \; a^{-1}$). At HEK, strain rates at most blocks in zone "A" in the central and orographic right part of the rock glacier surpassed this critical value for debris during the first half of the 2010s (exact years depend on the individual blocks, see Fig. 12).

Cicoira et al. (2020) and Bearzot et al. (2022) show that destabilization is characterized by discontinuities in BCF throughout the rock glacier area. They argue that such discontinuities in BCF and surface velocity are stronger indicators of destabilization than the magnitude of either parameter. HEK exhibits a pronounced pattern break just below the terrain step, where the destabilization - or "separation" as per Schneider and Schneider (2001) and Haeberli and Patzelt (1982) - of a part of the terminus is occurring (Fig. 8, Fig S2, supplement). In contrast, the upper section is much more homogenous. Compared to Plator rock glacier in Italy (Bearzot et al., 2022), HEK shows fewer distinct zones of similar BCF values but rather one main discontinuity.

Comparing absolute values of BCF at HEK to the larger dataset of Cicoira et al. (2020), we note that during highly destabilized conditions (phase 3), BCF at HEK as well as the corresponding velocities are higher than at any of the 340 rock glaciers (mostly in the French and Swiss Alps) assessed in their study. This could be explained by the difference between a landform-wide approach and the detailed investigations of single areas or even individual boulders (see also Bearzot et al. (2022)), and, in a more general sense, ties into the complexities of the dependence of spatial variability on scale. Under stable conditions, P1 and P3 have a BCF of about 1. In Cicoira et al. (2020), the rheological parameters were calibrated so that the peak of the distribution under normal conditions would match the value of 1. In this sense, HEK and its bulk creep behaviour are in line with most other rock glaciers analysed previously.

By extracting topographic information from the multi-temporal DSM dataset, we can extend the analysis presented in previous studies (Cicoira et al., 2020; Bearzot et al., 2022) and adjust the computation of the BCF for changes in slope angle and thickness over time. However, we do not account for changes in internal properties, i.e., friction angle, cohesion, shear resistance, etc., which can vary considerably over space and time and strongly depend on temperature (Cuffey and Paterson, 2010; Moore, 2014; Cicoira et al., 2019a; Millstein et al., 2022). The lack of detailed subsurface information limits the possibilities for modelling the movement of the rock glacier. We use the bedrock estimate by Hartl et al. (2016a) to calculate the BCF, which implies that we assume a unique shear horizon at the depth of the bedrock for the computation. In terms of rheology of perennially frozen materials in rock glaciers, this is a substantial assumption that likely deviates from reality in some if not all parts of the rock glacier. In general, using bulk parameters to describe rock glacier rheology - as in the computation of the BCF - is a strong simplification of complex processes and by definition does not resolve local variations in rock glacier composition, or external factors, which have to be taken into account by means of ad-hoc parameterisations. However, as shown, it is possible to use such bulk parameters for the definition of dynamic phases and to highlight different dynamic behaviours, which can then be investigated in more detail, especially when a multi-temporal, good quality data basis is available. Although the number of studies incorporating the relatively new concept of BCF into their analysis is limited, the correspondence between the HEK results from a geomorphological perspective and the more dynamic, BCF focussed approach are increasing our confidence in the validity of this method.

The geomorphological destabilization signs in combination with the analysis of elevation differences and pitch angles indicate a landslide-like behaviour with roto-translational kinematics. Decreasing surface elevation change at rock glaciers can typically be expected in cases of permafrost thawing and extensional flow. In contrast, positive elevation change is most often linked to the advance of a front, the re-stabilization of a scarp, or to compressive flow. The large elevation changes (> 2 m/a) observed during the destabilization phases at HEK appear to be related to the development of shearing surfaces and the consequent rotational and/or translational movement of the unstable permafrost masses. The change rates around the morphological destabilization features are orders of magnitude larger than in the area not affected by the destabilization process, and surface elevation change during highly destabilized conditions is generally higher than during the intermediate phase with normal creep behaviour. While change rates in the stable areas of HEK are in the range reported for similar landforms by other studies, the values in the destabilized section are considerably higher than at most other sites where such data is available (e.g. Kääb et al. (2003); Cusicanqui et al. (2021); Fleischer et al. (2021) and Wee and Delaloye (2022)). Elevation change rates similar to those in the destabilized section of HEK have recently been reported for the destabilized section of Tsarmine rock glacier in Switzerland (Vivero et al., 2022).

A detailed visual inspection of the DSMs and the hillshade maps reveals that the destabilization process develops from higher to lower elevations, indicating an enlarging landslide pattern. For both destabilization phases, the first morphological signs to appear are cracks at the sides of what will develop into a larger scarp. Other (subsequent) cracks appear further downslope as the ice-debris mixture loses cohesion, develops new sliding surfaces, and accelerates downslope. Springman et al. (2013) report a similar pattern as we observe in the HEK boulder rotations for Grabengufer rock glacier in Switzerland, where they found backwards tilting at the terminus and interpret this as evidence of "rotational failures at the foot of the rock

glacier combined with slumping". In one of the most comprehensive in situ investigations of rock glacier destabilization to date, Buchli et al. (2018) showed the existence of multiple shear horizons within the body of the Furggwanghorn rock glacier

(Switzerland) and that rotational as well as translational movement can be associated with individual shear zones. We consider our results to be additional supporting evidence of such processes, although further geomorphological analysis and detailed numerical modelling experiments are clearly needed for a more detailed assessment. Generally speaking, the evidence of rotational movement at destabilized rock glaciers suggests that the underlying processes may resemble slow moving landslides after rupture - past the tertiary creep phase - more closely than permafrost creep in rock glaciers. We hope the presented dataset

will contribute to advancing our process understanding in this direction.

### 4.3 Meteorological and climatological setting

Considering long-term changes, the last approximately three decades of profile velocities at HEK show two distinct peaks and subsequent slowdowns in 2004 and 2015-16, respectively. Nickus et al. (2015) suggest that the 2004 peak and subsequent deceleration at HEK may have been related to the very warm summer of 2003. Summer temperature anomalies in 2003 stand

out as an exceptionally warm outlier in the past two decades at the automatic weather station in Obergurgl (Fig. S7). 2015 was less extreme in terms of temperature anomalies but marks the start of a series of anomalously warm years that extends unbroken until 2022. The summers of 2018 and 2019 were the third and second warmest summers since 2003, respectively (2018 is tied with 2022 as the third warmest summer since 2003). 2015-2016, the years of the velocity plateau, saw comparatively dry summers. There was relatively little snow during the 2016/17 hydrological year but the summer of 2017 was unusually wet.

It is likely that high temperatures and liquid water input through precipitation both contributed to the renewed acceleration from 2017/18 onwards. However, rock glacier change monitoring at high temporal resolution in combination with modelling efforts is needed to gain a clearer picture of these connections. A number of rock glaciers in the Swiss, French, and Italian Alps also show a velocity peak in 2015 followed by 2-3 slower years and renewed acceleration from 2018 onwards, which strongly suggests a common climate forcing at the regional scale (Noetzli et al., 2019; Bearzot et al., 2022; Thibert and Bodin, 2022;

Wee and Delaloye, 2022).

The sub-seasonal TLS time series for the summer of 2019 primarily shows that movement rates at the front of the rock glacier fluctuate considerably on relatively short time scales, which is in keeping with other studies (e.g., Wirz et al. (2016); Buchli et al. (2018)). The magnitude of change per day (dividing total change by the length of the two week time span) is comparable to values reported for the front of Furggwanghorn rock glacier by Buchli et al. (2018). Short-term fluctuations of

730 change at the rock glacier front are likely related to meteorological forcing but the approximately two week resolution of the TLS data does not resolve potential correlations with individual precipitation events. Additionally, such correlations may be non-linear and not straightforward to identify. A speculative interpretation of the TLS data suggests that cumulative effects play a role for both temperature and precipitation on the time scales of the 2019 dataset. The first two week observation period (June 24 - July 6) was the warmest of the summer, but rock glacier movement was moderate. The third observation period, in which

maximum velocities were reached, was not as warm as the first, but saw a lot of precipitation and a significant temperature increase from close to freezing to about 15°C (Fig. S6, supplement). A similar jump in temperatures between periods 4 and 5

also coincides with an increase in velocity. Buchli et al. (2018) observed localized, high daily change rates in sections of the front that experience water outflows and link the occurrence of such outflows to hydrological processes, e.g. water flow due to snow melt. At HEK, large changes in temperature combined with liquid water in the rock glacier system seem plausible as general drivers of the short-term changes measured during the 2019 summer season, but continuous monitoring of movement and surface change is clearly needed to improve understanding of the rock glacier response to short-term meteorological input.

High-resolution monitoring is also needed to better assess the hazard potential of the destabilized section of the rock glacier. Localized rock fall from the steep rock glacier front has already led to temporary closures of the access road and is very likely to continue given the terrain and the movement of the front. The possibility of a complete collapse of the destabilized section of the rock glacier and a subsequent rapid mass movement cannot be ruled out given what we know about the internal and external contributing factors. However, large uncertainties remain about the composition of the rock glacier both in the lower, destabilized section as well as in the upper part. It is unclear how much material is affected by the destabilization, nor is it currently possible to predict the likelihood of a collapse.

## 5    Conclusions

We present an updated and extended dataset of in situ and remote sensing-based change monitoring from Äußeres Hochebenkar rock glacier consisting of 14 DSMs covering a time span of 68 years (Figure 2; photogrammetry using historical aerial imagery for 1953, 1971, 1977, 1990, and 1997, airborne laser scanning for 2006, 2009, 2010, 2011, and 2017, and UAV-borne laser scanning for 2018, 2019, 2020, and 2021), as well as 43 individual measurement years of block displacement on the rock glacier surface (starting in the early 1950s, annual resolution since 1997 with a one year gap in 2005 when no measurements were carried out), and a short-term time series of high-resolution 3D topographic change at the rock glacier front throughout the summer of 2019. Integrating the time series block displacement, DSM derived velocities, and geomorphological analysis allows for a detailed assessment of two cycles of destabilization at HEK. While recent velocities measured in the lower sector of the rock glacier are unprecedented in the long time series, the destabilization process driving these kinematic changes is not. The highest velocities were recorded in the most recent measurement years and reach values > 23 m/a (DGNSS, single blocks, 2021/22) and about 30 m/a (image correlation from ULS data, 2020/21) in the highly destabilized lower section of the rock glacier.

Combining in situ data of block displacement and high-resolution 3D point clouds allows for method development based on data fusion approaches by expanding the change detection capacity of the monitoring network to new parameters, such as block rotation. Interdisciplinary monitoring and the coordinated integration of data and methods have the potential to shed light on the processes controlling rock glacier dynamics through geomorphological analysis and numerical modelling based on the resulting data. Such advances are only practicable when the data basis shows high granularity in terms of temporal and spatial resolution as well as accuracy.

We wish to highlight the importance of maintaining the monitoring network at Äußeres Hochebenkar rock glacier and, as a future goal, expanding it to subsurface monitoring. Well-studied sites like HEK are essential for the development and testing

of new methods, help with determining the most suitable monitoring approaches for newly established sites, and contribute to advances in fundamental process understanding. The extensive historical data basis extending nearly 70 years into the past and the cyclic destabilization behaviour of the rock glacier provide a promising starting point for the development, calibration, and validation of numerical models pertaining to rock glacier movement in general and the destabilization process in particular. We are confident the dataset presented in this study (freely available through data repositories, see section Data Availability) will

contribute to such efforts.

*Code and data availability.*     – The DGNSS time series of block velocities is available on the pangaea data repository and updated yearly (Stocker-Waldhuber et al., 2021).

  – The multi-temporal digital surface models, shaded reliefs and differential digital surface models as well as the derived velocity vectors are available on the Zenodo data repository (Zieher et al., 2022).

– The TLS data for summer 2019 and the source code for the correspondence-driven plane-based M3C2 method are available at: Zahs et al. (2021).

  – The code to derive the velocity vectors and for the visualisations in this paper can be found at https://github.com/thomaszieher/HRG_reanalysis

  – The code to make the figures in this manuscript can be found at: https://github.com/LeaHartl/Hochebenkar_figures

*Video supplement.* An animated version of the time series of DSM hillshades is available at https://av.tib.eu/media/60175 (Cicoira et al., 785 2022)

*Author contributions.* TZ, MB, and CK generated the DSM time series from source data. TZ homogenised the DSM dataset and derived the velocity time series. MB carried out the analysis related to the rotational movement of blocks. MS provided data and analysis related to the DGNSS block displacement and coordinates the in situ monitoring program. VZ and BH provided TLS data of the rock glacier front and carried out detailed 3D change analysis based on this data. LH and AC conceptualised the paper and carried out the kinematic and 790 geomorphological analyses of the time series. LH wrote the paper with contributions from all co-authors.

*Competing interests.* The authors declare that they have no conflict of interest.

*Acknowledgements.* The contributions of Heralt and Britta Schneider to the historical time series of block displacement are invaluable. In recent times, the maintenance of the block displacement time series and related monitoring efforts are supported by the "Verein Gletscher und Klima" and carried out by a team consisting mostly of volunteers. Without their dedication, continued long-term monitoring would not 795 be possible.

Open source software was essential to the data processing, analyses, and visualizations in this study, in particular: QGIS, SAGA, R (R Core Team, 2021), and numerous python packages (QGIS.org, 2010; Oliphant, 2006; Hunter, 2007; GeoPandas, 2013-2019; Gillies et al., 2013; Conrad et al., 2015; Harris et al., 2020).

We are very grateful to Wilfried Haeberli and an anonymous reviewer for their constructive and thoughtful comments. We appreciate the time spent on this manuscript by the reviewers and the editorial team and sincerely thank them for their help!

800

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
