# Peer review of "Multisensor monitoring and data integration reveal cyclical destabilization of Äußeres Hochebenkar Rock Glacier"

_Earth Surface Dynamics, 2022_

## Referee Comment (RC2)

[referee-annotated manuscript omitted]

---

## Author Comment (AC1)

Thank you for these very encouraging comments! We appreciate the time spent on the review and will implement all the suggestions to the best of our ability in a revised version of the manuscript. We respond to specific comments below.

For readability, reviewer comments are in black in this document and our responses are in blue.

**General comments**

The spectacular active rock glacier in the Äusseres Hochebenkar of the Ötztal Alps, Austria, is among the best documented viscous creep features in warm and warming mountain permafrost. The submitted paper compiles, homogenizes and interprets geodetic/photogrammetric observations of extraordinary length (68 years) complemented by modern airborn-uncrewed laser scanning with extreme spatio-temporal resolution. Emphasis is on technical innovation and on the analysis of flow acceleration to destabilization. The unique and extraordinarily rich data set including such novel aspects as, for instance, subseasonal effects and phenomena of block rotation enables detailed insights concerning the landslide-type evolution of the steep frontal part with an earlier, weaker destabilization between the 1950s and 1970s, followed by more stable conditions until the most recent, ongoing and extreme acceleration and destabilization. The text is well written and illustrated, follows a clear/logical structure and discusses the applied techniques and observed phenomena at the forefront of the rapidly growing research field and scientific literature. The paper can essentially be recommended for publication in its present form. The authors may, nevertheless, wish to consider the following reflections.

More could be said about local permafrost conditions as the decisive environmental aspect related to the analyzed creep phenomena. The high-resolution, Alpine-wide permafrost map by Boeckli et al. (2012) should at least be mentioned. A superposition using the publicly available kmz file of permafrost occurrence onto a Google-Earth image would be instructive (Figure 1 in Haeberli 2013 is an example from the Oetztal region).

**We will add the permafrost map to Fig. 1 in the manuscript as suggested and expand comments on the permafrost conditions in the study site in the text.**

The excellent meteo-data in the region can be used to calculate mean annual air temperatures at the investigated site. This would immediately make clear that permafrost is quite warm despite the cooling effects of the ventilated surface layer consisting of coarse blocks.

We will add more detail on the climatological conditions based on the meteo-data and previous publications.

Mention ongoing permafrost warming trends as documented by borehole measurements (Etzelmüller et al. 2020) and related paleo-effects at depth (thermal anomaly down to >

50m). The perennially frozen, ice-rich condition of the creeping mass is documented by high but variable P-wave velocities and by electrical D.C. resistivities in the medium to high kOhmm range.

**We will include a reference to Etzelmüller et al. 2020 and give context in the text, as suggested.**

The geometry and kinematics should strictly relate to the moving mass of perennially frozen talus rather than to the landform "rock glacier" (as mental constructs and conventions, landforms can – strictly speaking – not flow). It is important to make clear that the thickness of the landform rock glacier (sediment above bedrock?) can differ from the depth of the thermally defined permafrost, and that the thickness of the moving mass may depend on the occurrence of shear horizons rather than on landform thickness or permafrost depth.

**We agree with the reviewer that this is an important issue and will revise the text to improve clarity throughout the different sections.**

While the term "acceleration" is self-explaining, the term "destabilization" is more complex and needs a precise definition. Critical strain rates for crevasse formation as an indication of tensile strengths had already been discussed at Gruben rock glacier (Haeberli et al 1979) and may relate to the onset of rapid local sliding. Interesting destabilization phenomena have also been documented in creeping subarctic permafrost (Daanen et al. 2012).

We will add more context regarding the term and concept of "destabilization" as we use it here, based on the suggested references.

**Minor remarks**

Line 25: In connection with the fully justified statement that "... rock glaciers are ... generated by ... creep of frozen ground ...", the reference to the PhD thesis of Whalley (1974) is astonishing. Whalley had postulated that rock glaciers are debris-covered glaciers and that permafrost is unlikely to occur in the Alps. Still today and against all measured evidence, this author keeps fighting against what he calls "the permafrost model" of rock glacier formation (see the recent discussion in The Cryosphere). The authors should either mention this fact or skip this reference. The latter may be more adequate as long-outdated beliefs from intuitive landform interpretation are at best of historical interest and hardly relate to the excellent quantitative material presented in the submitted paper.

**We will remove this reference.**

Line 66: The paper by Daanen et al. about permafrost destabilization in the Brooks Range could be cited here.

**Will add citation of Daanen et al.'s work on the Dalton Highway debris lobes.**

Line 100: Zahs et al. (2019) report resistivities (medium to high kOhmm range) at the rock glacier margin which indicate ice-rich frozen ground with variable ice content, not just "isolated ice-lenses").

**We will adjust the text to better reflect this.**

Table 1: Take care of "Umlaute" and be consistent: Ladstädter/Ladstaedter. Check throughout the paper and the reference list.

Will check for consistency of Umlaute.

Figure 1: The top-left graph could include the regional permafrost occurrence after Boeckli et al. (2012). The longitudinal profile and the central flowline in the lower-rigth graph cannot easily be discriminated and are not identical – flow directions in the root zone deviate from the given black line.

We will include the permafrost map and attempt to improve the visualization of the flow line.

Bulk creep factor (BCF): This is a useful but rather abstract concept. At an individual location, where surface slope remains nearly constant and changes in flow depth cannot realistically be accounted for, changes in BCF directly reflect changes in surface velocities. The authors rightly emphasize that changes in space and time of both, BCF and strain rates combined, may most significantly indicate the onset of rapid sliding-type movements.

**Will edit text to emphasize this more strongly.**

Line 279: Be more precise concerning depth values for landforms, permafrost and flowing mass as explained above.

We will revise text to ensure consistent and clear terminology.

Line 296: Perhaps better " the frozen mass of the rock glacier entered ... "

Agreed.

Line 381: what does "often by a substantial margin" mean?

Will edit text to be more precise.

Figure 8: The important information is hard to deciffer – enlarge.

We will rework the figure to improve this.

Lines 400-406: Be consistent with the times (present – past) used in writing.

We will revise the manuscript to ensure consistent tenses.

Line 415: These values may be compared with the "critical strain rates for crevasse formation" discussed for Gruben rock glacier by Haeberli et al.(1979).

Will add citation and compare values.

Line 568: Perhaps better "rheology of perennially frozen materials in rock glaciers"

Line 598: Perhaps better "permafrost creep" or "premafrost creep in rock glaciers"

Agreed.

Line 821: Check "Umlaute": Blockströmen, Ötztaler

Will check for consistent Umlaute.

I congratulate the authors for their outstanding work and hope their important contribution to be published soon.

Thank you again for this very encouraging and motivating review!

---

## Author Comment (AC2)

We thank the reviewer for the constructive comments and the time spent on the manuscript! We will implement the suggestions to the best of our abilities. In the following, responses to specific comments are in blue text. Reviewer comments are black.

**Structure:** The paper is based on data and methods presented in previous studies of the authors. Without a knowledge of these studies, it is sometimes hard to follow.**

We will revise the relevant sections to improve clarity. We aim for a balance between showing new work and combining it with data and results from previous studies, as this combination leads to new insights and a more comprehensive understanding of the rock glacier. We agree that enough information must be given for the reader to follow and will work to improve this issue.

The methods can be described in a more comprehensive way. Data acquisition and analyses methods are mixed in section 2. The methods (e.g. image correlation or accuracy assessment) are described in sentences which go very deep into the details but are not comprehensible and useful to understand the method. If you use such a deep level of description, you must explain much more of the method. As alternative, use a simple way to describe the method in a few sentences and give a overview.

We accept this criticism and will separate data and acquisition more clearly in the revised manuscript. We will also reconsider the level of detail and adjust the explanation appropriately. Thank you for pointing this out.

In the discussion, topics are discussed which are not described in the method and data section. Please adjust method and discussion section.

**We will make sure all topics relevant to the discussion are included in the methods section.**

**Figures:** The selection of the figures should be revised. It is confusing referring in the text to figures from the supplement. Figures in the manuscript are too small and it is hard to get the information from it.

We will revise the figure selection and increase the size to improve readability. In particular, we will add figures pertaining to the uncertainty analysis to the manuscript. These were in the supplementary material previously but can be incorporated in the main publication to improve clarity.

**Language:** The paper should be also revised in terms of sentence structure. In some parts sentences are very long and hard to follow.

We will revise the language and take care to shorten sentences where necessary.

**Conclusion:** How can you know that the onset of destabilization was in 2017 when you do have no data between 2011 and 2016?

We do have block displacement data for the 4 block profiles for this time period at annual resolution, as well as strain rates computed from the displacement data. We will rephrase the relevant sections to make this more clear. We acknowledge that the term "destabilization" requires careful explanation and clarity in terms of definitions and usage and will work to improve this in a revised version of the manuscript.

**Specific comments - page and line numbers refer to the annotated pdf uploaded with the review.**

P2: check sentence structure for this chapter. Very long sentences are hard to follow and not always correct

We will revise the language in this chapter to improve sentence structure.

**P3: L58 Crystalline lithology: and in what else?**

Marcer et al. (2019) find destabilizing rock glaciers in densely jointed lithologies (i.e. ophiolites and schists) as opposed to crystalline lithologies. We will add this information to the text. P3, additional comments:

Will fix typo and adjust wording.

P4: sliding? - yes, we will change it.

P5: lidar in 1953?

No, DSMs are derived from aerial imagery for the early data sets. Will rephrase for clarity.

**P8: new section=> this is no geodetic monitoring**

**Will change as suggested.**

The data acquisition is described in a chaotic way. Information about data acquisition and methods are mixed and described at very different levels of detail. It is hard to follow which sensors and data where used. Revise this. Further a table with the datasets and ist accuracy is missing

We will rework this section to improve the issues mentioned here.

**Using which software?**

The ICP algorithm was used as a plug-in of the FOSS SAGA GIS (Conrad et al. 2015), which is implemented in c++. The implementation was done by the authors and is an adaptation of the original algorithm presented by Besl & McKay (1992). We will clarify this in the text.

**DSM as abbreviation is already introduced**

Yes, we will adjust the text and fix the typo/missing word in the previous sentence. We will also revise the sentence structure in the following lines.

**P9: without a knowledge of the image correlation tool it is very hard to follow. You write about DSM, grids and now points. Revise this description**

We will revise this section and the following paragraphs to improve clarity. We will add a conceptual figure to explain the employed correlation tool, together with a more comprehensive description.

P10: Why have you chosen this method for uncertainties assessment. Add references with a statistic legitimization for this method?

We will add a more detailed explanation and references for the chosen method.

the ULS campaign is described in detail while no information about the TLS campaign. Why? The TLS campaigns have been described in previous publications (cited in Table 2), while the ULS campaigns have not. We will make this more clear in the text and add the citations for the TLS campaigns to this section. We will revise sentence structure and phrasing in the last paragraph on P10 as indicated by the reviewer in the pdf.

**P11: you used orthoimages? First time you mention. I miss this information in the previous section**

We will add this to the previous section. We mention the ortho images in table 2 but agree with the reviewer that this needs an explanation in the text.

L255 x y z?: We will rephrase for clarity. This led to a 3D translation vector (x,y,z) L257: Implementation in FOSS SAGA GIS by the authors in c++ programming language, as above.

**Table 2**

We will adjust the table headings and content of the table as suggested.

P15 - we will revise the sentence structure where marked by the reviewer

**L299 why is that?**

Poor quality of the underlying data (aerial images) in the area of the terminus. We will add this explanation to the text.

**L305 why?**

There are no block profiles in the lowest part of the terminus, hence the block profiles do not capture the change of this section. Will clarify in the text.

geomorphological analyses should appear somewhere in the method section

Agreed. We will incorporate these analyses in the methods section.

**P16: now you mention elevation differences. Describe the difference methods more clearly in the method section**

Will add a better description in the methods section.

**L320 ???geomorphological features?**

Yes, we refer to geomorphological destabilization features such as crevasses, cracks, and scarps. We will add a clearer definition of the term "destabilization signs" and incorporate this in the updated methods section on the geomorphological analyses.

**L334: why? these has to be mentioned in the methods section!**

The data quality of the first five DSMs is not as good as later on due to the quality of the aerial imagery used to derive these DSMs. The DSMs computed from the available digitized analogue images do not include the same level of topographic detail as the DSMs derived from laser scanning, which is an active remote sensing technique. We will add an explanation to clarify this.

**P18 - will fix typo**

**Figure 5 it is easier to mark geomorphological features by a line and not a letter!**

We would like the features to remain visible to the reader in the images and would prefer not to mark them with lines, as this would obscure the complexity of the features. We also note that we

use the letters to refer to zones of the rock glacier, rather than individual geomorphological features. We will experiment with different options for this figure to try and improve clarity.

**Figure 8: figure is too small and hard to read**

Will increase size of the figure.

**P22 L391 ff what do you mean?**

We will rephrase to improve clarity. The image correlation technique is applied at regularly spaced nodes (raster cells) in a distance of 5m. Applying the technique at each node wouldn't be possible from a computational perspective. However, the nodes left out could provide slightly different results compared to the closest considered node. The image correlation method will be described in more detail in the methods section.

**P23 why now named as profiles 0, .. and not P0 ?**

**Will change to P0 and check consistency.**

P25: L439 when? Timespans I-V in summer 2019, as shown in Fig. 12. Will add this to the text. The discussion about the uncertainties in the first part of the chapter is important but unfortunately are the methods and results not presented in the previous chapters. I am missing a structure e.g. chronological order of the datasets

We will rewrite this section to improve clarity, structure, and completeness. We will also ensure that all relevant topics discussed in this section are mentioned in the methods section.

**P26 L452 what does this mean?**

Will add detail to improve clarity, see also the response to comment on P22, L391.

**L454 but this is not presented in this study.**

We will rework this section to improve clarity. The co-registration of the multi-temporal datasets as discussed here will be better described in the methods section. We agree with the comment about the presentation, the boxplots originally presented in the supplement will be presented in the text as an additional and revised figure.

**L460 showing the uncertainties in m/a makes it difficult to compare the quality of the datasets. Using the absolute value is more suited when discussing the data quality**

In the light of the scope of the analysis - an assessment of velocity changes - we respectfully disagree with this statement. The periods between subsequent epochs are irregular, which would make it more tricky to compare derived distances directly. After normalizing the distances by the time periods, the mean annual velocities and their velocities become directly comparable. We will add an explanation of our reasoning to the manuscript and more clearly include references to publications that go into greater detail on absolute uncertainty measures of the previously published data sets (these references are currently cited in Table 2).

**L 464 if you discuss this you have to mention it in the method section!**

This refers to work presented in previous studies. We will clarify this and add the relevant reference again here.

P28: a comprehensive table or figure would make it easier to follow the time series

We agree that such an overview would be helpful to the reader. We will add either a figure or revise table 2, depending on whether we can find a readable way to visualize the data sets throughout the time series.

P28 add the date: 1953-2006, will add this in the text.

**P29 how can you know that the onset of destabilization was in 2017 when you do have no data between 2011 and 2016?**

We do have the annual block displacement data from the 4 block profiles. The statement is based on destabilization signs visible in the 2017 DSM and these data. We will rephrase this section for clarity.

**which region?**

The majority is in the French and Swiss Alps, a smaller number from the Austrian and Italian Alps. Will add this in the text.

**P31: Subsequent?? - yes, will adjust this.**

in the supplement only summer precipitation is shown. What is with snow accumulation? 2017 was little in snow

We will adjust the figure to include winter precipitation / snow, and add a note on this in the text. ???

Typo, will adjust.

**P32: since about?**

Will change to: "annual since 1997 with the exception of 2005, when no measurement took place."

**hard to read**

Will rephrase to improve clarity.

Thank you again for the detailed review and thoughtful comments!

**References**

Besl, P. J. and McKay, N. D.: Method for registration of 3-D shapes, in: Sensor fusion IV: control paradigms and data structures, vol. 1611, pp. 586–606, Spie, 1992.

Conrad, O., Bechtel, B., Bock, M., Dietrich, H., Fischer, E., Gerlitz, L., Wehberg, J., Wichmann, V., and Böhner, J.: System for automated geoscientific analyses (SAGA) v. 2.1. 4, Geoscientific Model Development, 8, 1991–2007, 2015.

Marcer, M., Serrano, C., Brenning, A., Bodin, X., Goetz, J., and Schoeneich, P.: Evaluating the destabilization susceptibility of active rock glaciers in the French Alps, The Cryosphere, 13, 141–155, 2019.

---

## Author Response (AR1)

**Summary of changes made to the manuscript:**

We have adapted the manuscript to incorporate the suggestions by both reviewers. The main changes in the revised version of the manuscript are related to the structure of the methods and discussion sections, as well as the figure selection. In particular, we have:

- Added an overlay of the permafrost index map of Boeckli et al (2012) to Fig 1
- Added a figure showing the availability of different data sets over time (new Fig 2)
- Added a conceptual figure to explain the uncertainty assessment (new Fig 3)
- Redesigned a figure previously contained in the supplement and added it to the discussion section to illustrate the uncertainty assessment (new Fig 15)
- Changed the structure of the methods section to better explain the uncertainty assessment and more clearly differentiate between new data and analyses and previous work
- Added subsections in the methods for meteorological data and geomorphological analysis
- Expanded and restructured the discussion to improve the explanation of the uncertainty assessment

**For detailed responses to specific reviewer comments please see the following pages of this document.**

We also want to point out that we have added data from the in situ monitoring of block displacement from measurements carried out in summer 2022. This adds an extra data point to the block displacement time series but does not change the overall interpretation or conclusions. The 2022 data was not yet available when we first submitted this manuscript.

Thank you for your time and consideration!

**Response Review 1**

Thank you for these very encouraging comments! We appreciate the time spent on the review and will implement all the suggestions to the best of our ability in a revised version of the manuscript. We respond to specific comments below.

For readability, reviewer comments are in black in this document and our responses are in green.

**General comments**

The spectacular active rock glacier in the Äusseres Hochebenkar of the Ötztal Alps, Austria, is among the best documented viscous creep features in warm and warming mountain permafrost. The submitted paper compiles, homogenizes and interprets geodetic/photogrammetric observations of extraordinary length (68 years) complemented by modern airborn-uncrewed laser scanning with extreme spatio-temporal resolution. Emphasis is on technical innovation and on the analysis of flow acceleration to destabilization. The unique and extraordinarily rich data set including such novel aspects as, for instance, subseasonal effects and phenomena of block rotation enables detailed insights concerning the landslide-type evolution of the steep frontal part with an earlier, weaker destabilization between the 1950s and 1970s, followed by more stable conditions until the most recent, ongoing and extreme acceleration and destabilization. The text is well written and illustrated, follows a clear/logical structure and discusses the applied techniques and observed phenomena at the forefront of the rapidly growing research field and scientific literature. The paper can essentially be recommended for publication in its present form. The authors may, nevertheless, wish to consider the following reflections.

More could be said about local permafrost conditions as the decisive environmental aspect related to the analyzed creep phenomena. The high-resolution, Alpine-wide permafrost map by Boeckli et al. (2012) should at least be mentioned. A superposition using the publicly available kmz file of permafrost occurrence onto a Google-Earth image would be instructive (Figure 1 in Haeberli 2013 is an example from the Oetztal region).

**We added an overlay of the permafrost map to Fig. 1 as suggested, as well as a citation of Boeckli et al. (2012) and note on this in the text.**

The excellent meteo-data in the region can be used to calculate mean annual air temperatures at the investigated site. This would immediately make clear that permafrost is quite warm despite the cooling effects of the ventilated surface layer consisting of coarse blocks.

Added a paragraph to the introduction giving MAAT in Obergurgl for the climatological reference periods 1961-90 and 1991-2020 and pointed out that MAAT at the rock glacier is near 0°C based on the shorter time series from the AWS at the rock glacier.

Mention ongoing permafrost warming trends as documented by borehole measurements (Etzelmüller et al. 2020) and related paleo-effects at depth (thermal anomaly down to > 50m). The perennially frozen, ice-rich condition of the creeping mass is documented by high but variable P-wave velocities and by electrical D.C. resistivities in the medium to high kOhmm range.

**We have added a reference to Etzelmüller et al. 2020 in the introduction to contextualize the degradation of permafrost at the study site.**

The geometry and kinematics should strictly relate to the moving mass of perennially frozen talus rather than to the landform "rock glacier" (as mental constructs and conventions, landforms can – strictly speaking – not flow). It is important to make clear that the thickness of the landform rock glacier (sediment above bedrock?) can differ from the depth of the thermally defined permafrost, and that the thickness of the moving mass may depend on the occurrence of shear horizons rather than on landform thickness or permafrost depth.

**We have revised the manuscript to more consistently differentiate between the landform and the moving mass, using the terminology suggested above.**

While the term "acceleration" is self-explaining, the term "destabilization" is more complex and needs a precise definition. Critical strain rates for crevasse formation as an indication of tensile strengths had already been discussed at Gruben rock glacier (Haeberli et al 1979) and may relate to the onset of rapid local sliding. Interesting destabilization phenomena have also been documented in creeping subarctic permafrost (Daanen et al. 2012).

**We have added a reference to Daanen et al. 2012 in the introduction. In the discussion we now include a comparison of our findings on strain rates to those of Haeberli et al. 1979.**

**Minor remarks**

Line 25: In connection with the fully justified statement that "... rock glaciers are ... generated by ... creep of frozen ground ...", the reference to the PhD thesis of Whalley (1974) is astonishing. Whalley had postulated that rock glaciers are debris-covered glaciers and that permafrost is unlikely to occur in the Alps. Still today and against all measured evidence, this author keeps fighting against what he calls "the permafrost model" of rock glacier formation (see the recent discussion in The Cryosphere). The authors should either mention this fact or skip this reference. The latter may be more adequate as long-outdated beliefs from intuitive landform interpretation are at best of historical interest and hardly relate to the excellent quantitative material presented in the submitted paper.

**We removed the reference.**

Line 66: The paper by Daanen et al. about permafrost destabilization in the Brooks Range could be cited here.

**Added citation of Daanen et al. (2012).**

Line 100: Zahs et al. (2019) report resistivities (medium to high kOhmm range) at the rock glacier margin which indicate ice-rich frozen ground with variable ice content, not just "isolated ice-lenses").

We adjusted the text to better reflect this, the section now reads:

"Zahs et al. (2019) found no permafrost in an electrical resistivity tomography (ERT) profile beside the rock glacier at an elevation of about 2470 m. In a profile at similar elevation on the rock glacier terminus they reported resistivities indicative of ice-rich frozen ground with variable ice content."

Table 1: Take care of "Umlaute" and be consistent: Ladstädter/Ladstaedter. Check throughout the paper and the reference list.

**Checked manuscript for consistency of Umlaute/spelling of this name.**

Figure 1: The top-left graph could include the regional permafrost occurrence after Boeckli et al. (2012). The longitudinal profile and the central flowline in the lower-rigth graph cannot easily be discriminated and are not identical – flow directions in the root zone deviate from the given black line.

**We added the permafrost map of Boeckli et al. (2012) as suggested and changed the line style of the flowline to hopefully make it more visible.**

Bulk creep factor (BCF): This is a useful but rather abstract concept. At an individual location, where surface slope remains nearly constant and changes in flow depth cannot realistically be accounted for, changes in BCF directly reflect changes in surface velocities. The authors rightly emphasize that changes in space and time of both, BCF and strain rates combined, may most significantly indicate the onset of rapid sliding-type movements.

Added a sentence to emphasize this more strongly in the subsection of the methods on BCF: "In combination with changes in surface strain rates, spatial and temporal patterns in surface velocities, and morphological destabilization signs, discontinuities in BCF provide a further indication of the onset of rapid sliding-type movement."

Line 279: Be more precise concerning depth values for landforms, permafrost and flowing mass as explained above.

**Edited this section as follows:**

"Rock glacier thickness is given by a map of the rock glacier's bedrock extrapolated from GPR data and presented in Hartl et al. (2016). This is a simplification as the depth of the bedrock may differ substantially from the depth of the thermally defined permafrost and the

thickness of the moving mass. Nonetheless, lacking more detailed subsurface information on layering and potential shear horizons, we consider the approximate bedrock depth the best available estimate for our application."

Line 296: Perhaps better " the frozen mass of the rock glacier entered ... "

Changed as suggested.

Line 381: what does "often by a substantial margin" mean?

Text now reads as follows (exact values for each year can be found in Fig. 4 and the data publication): "Since its establishment in 1997, P0 was the slowest of the 4 profiles, suggesting advanced degradation of the rock glacier in the terminus area. P3, the next slowest profile, was typically faster than P0 by 0.2m/a or more."

Figure 8: The important information is hard to deciffer – enlarge.

Changed layout of figure to improve readability.

Lines 400-406: Be consistent with the times (present – past) used in writing.

Adjusted text for consistent use of the past tense in this section.

Line 415: These values may be compared with the "critical strain rates for crevasse formation" discussed for Gruben rock glacier by Haeberli et al.(1979).

Added citation and comparison of values in the discussion section. New text:

"In a study assessing the surface movement of Gruben rock glacier (CH) Haeberli et al. (1979) suggest that the critical strain rate for the formation of crevasses is lower for frozen debris material ( $2.7 \pm 0.9 \ 10 \ -3 \ a-1$ ) than for massive ice ( $1.4 \pm 1.0 \ 10-2 \ a-1$ ). At HEK, strain rates at most blocks in Zone "A" in the central and orographic right part of the rock glacier surpassed the critical value for debris during the first half of the 2010s (exact years depend on the individual blocks, see Fig. 11)."

Line 568: Perhaps better "rheology of perennially frozen materials in rock glaciers"

**Changed as suggested.**

Line 598: Perhaps better "permafrost creep" or "premafrost creep in rock glaciers"

Changed to "permafrost creep in rock glaciers"

Line 821: Check "Umlaute": Blockströmen, Ötztaler

Fixed Umlaute.

I congratulate the authors for their outstanding work and hope their important contribution to be published soon.

Thank you again for this very encouraging and motivating review!

\_\_\_\_\_

**Response Review 2**

We thank the reviewer for the constructive comments and the time spent on the manuscript! We will implement the suggestions to the best of our abilities. In the following, responses to specific comments are in green text. Reviewer comments are black.

**Structure:** The paper is based on data and methods presented in previous studies of the authors. Without a knowledge of these studies, it is sometimes hard to follow. We have revised the manuscript to improve clarity. We aim for a balance between showing new work and combining it with data and results from previous studies, as this combination leads to new insights and a more comprehensive understanding of the rock glacier. We agree that enough information must be given for the reader to follow and have worked to improve this issue with additional figures (e.g., figures showing timeline of data availability) and an updated

section and subsection structure.

The methods can be described in a more comprehensive way. Data acquisition and analyses methods are mixed in section 2. The methods (e.g. image correlation or accuracy assessment) are described in sentences which go very deep into the details but are not comprehensible and useful to understand the method. If you use such a deep level of description, you must explain much more of the method. As alternative, use a simple way to describe the method in a few sentences and give a overview.

We have restructured the methods and data section to more clearly differentiate between previously published data and new data and analyses. Our study is based on a combination of previous results, new data, and a homogenisation of these time series. For the sake of brevity, we refer to existing publications for in depth details on acquisition and processing of older data sets. We describe data and methods in more detail for the data and analyses that have not been published previously. We have revised the manuscript to make it clear where we are describing new data, where we are describing older data which have been covered in detail by previous work, and where we describe the methods we use to homogenise and combine the two.

In the discussion, topics are discussed which are not described in the method and data section. Please adjust method and discussion section.

We have restructured the methods and data section and the discussion to ensure completeness. Please see the responses to the specific comments below for further details.

**Figures:** The selection of the figures should be revised. It is confusing referring in the text to figures from the supplement. Figures in the manuscript are too small and it is hard to get the information from it.

We have revised the figure selection and increased the size of small figures to improve readability. We have added:

- Conceptual figure in the methods section to explain the principles of our uncertainty analysis
- Figure showing uncertainties in North-South, East-West, and vertical direction in the discussion (a version of this was previously included in the supplementary material)
- Overview figure showing timeline of data availability

**Language:** The paper should be also revised in terms of sentence structure. In some parts sentences are very long and hard to follow.

We have revised the sentence structure throughout the manuscript to improve readability.

**Conclusion:** How can you know that the onset of destabilization was in 2017 when you do have no data between 2011 and 2016?

This statement is based on destabilization signs visible in the 2017 DSM. We do have additional annual block displacement data from the 4 block profiles for 2011-2016. We have rephrased the sentence to make clear what we are referring to and to better express the ambiguity associated with the lack of DSMs between 2011 and 2017. We acknowledge that the term "destabilization" requires careful explanation and clarity in terms of definitions and usage and have tried to improve this throughout the revised version of the manuscript.

**Specific comments - page and line numbers refer to the annotated pdf uploaded with the review.**

P2: check sentence structure for this chapter. Very long sentences are hard to follow and not always correct

We have revised this section to improve clarity and sentence structure.

**P3: L58 Crystalline lithology: and in what else?**

Marcer et al. (2019) find destabilizing rock glaciers in densely jointed lithologies (i.e. ophiolites and schists). We have added this to the text - revised sentence:

"Interestingly, they note that destabilizing rock glaciers tend to be pebbly as opposed to bouldery (Ikeda and Matsuoka, 2006) and in densely jointed lithologies (i.e. ophiolites and schists) as opposed to crystalline lithologies."

**P3, additional comments:**

Fixed typo, adjusted wording. P4: sliding? - yes, changed to sliding P5: lidar in 1953? No, DSMs are derived from aerial imagery for the early data sets. Rephrased for clarity.

P8: new section=> this is no geodetic monitoring Added new section heading for the meteorological data.

The data acquisition is described in a chaotic way. Information about data acquisition and methods are mixed and described at very different levels of detail. It is hard to follow which sensors and data where used. Revise this. Further a table with the datasets and ist accuracy is missing

We have restructured this section to improve the issues mentioned here. The section is now more clearly divided into subsections addressing 1) recent high resolution monitoring (ULS, TLS) 2) the longer DSM time series and the derived time series of surface velocity and elevation change, and 3) rotational information for individual blocks derived from ULS point clouds and in situ data. We have adapted Table 2 based on the reviewer suggestions and more clearly refer to previous publications for detailed information on the accuracy of older data sets.

**Using which software?**

The ICP algorithm was used as a plug-in of the FOSS SAGA GIS (Conrad et al. 2015), which is implemented in c++. The implementation was done by the authors and is an adaptation of the original algorithm presented by Besl & McKay (1992). We have added this information to the text.

**DSM as abbreviation is already introduced**

We fixed this as suggested and revised sentence structure where indicated by the reviewer as part of a general restructuring of the section.

**P9: without a knowledge of the image correlation tool it is very hard to follow. You write about DSM, grids and now points. Revise this description**

We have rewritten and restructured this section to improve clarity. We have also added a conceptual figure to explain the correlation tool.

**P10: Why have you chosen this method for uncertainties assessment. Add references with a statistic legitimization for this method?**

We have revised this for improved clarity. The revised text now reads: "In order to show patterns of elevation gain and loss, differential digital surface models (DDSMs) were computed by subtracting the DSMs of two consecutive epochs (Williams, 2012). DDSM uncertainty was assessed by computing the 2.5% and 97.5% quantile of the elevation difference within stable bedrock outcrops. This provides an estimate of the inherent noise and, hence, the detection threshold for obtaining significant surface changes (Table 2) (Williams, 2012)."

**the ULS campaign is described in detail while no information about the TLS campaign. Why? The TLS campaigns have been described in previous publications (cited in Table 2), while the ULS campaigns have not. We have rephrased the text to make this more clear. (New text: "...This temporally highly resolved dataset complements an annual TLS time series starting in 2015 and described in Ulrich et al. (2021) and Zahs et al. (2022a). We refer to these**

publications for more detailed information on the TLS data acquisition and measurement set up.")

We also revised the sentence structure in the section on the CD-PB M3C2 algorithm, as indicated by the reviewer in the pdf.

**P11: you used orthoimages? First time you mention. I miss this information in the previous section**

Yes, orthoimages were generated as part of the ULS campaigns (mentioned in Table 2). We have added a note on this to the previous section (Subsection "Recent high-resolution of 3D topographic change").

L255 x y z?: Rephrased for clarity. New sentence: "This led to a 3D translation vector (x, y, z) describing the estimated block movement between two epochs."

L257: Implementation in FOSS SAGA GIS by the authors in c++ programming language, as above.

**Table 2**

Adjusted table headings as suggested, rephrased for clarity

P15 - Revised sentence structure where marked by the reviewer.

**L299 why is that?**

Shading problems in the underlying data (aerial images) in the area of the terminus. We have added this explanation to the text.

Revised text: "It should be noted that this DSM pair does not resolve the terminus well due to shading effects in the underlying aerial imagery. Hence, the DSM pair for this epoch likely does not capture the full range of velocities in the lowest part of the rock glacier."

**L305 why?**

There are no block profiles in the lowest part of the terminus, hence the block profiles do not capture the change of this section.

Revised text: "There was no block profile in the lowest section of the terminus at this time, so the DSM-derived velocity vectors show processes at the terminus that were not captured by the in-situ monitoring."

geomorphological analyses should appear somewhere in the method section. We have added a subsection on this to the methods section.

**P16: now you mention elevation differences. Describe the difference methods more clearly in the method section**

We added a clearer and more extensive description of this in the methods section (Subsection "Long-term change monitoring"). The elevation differences are a result of the DDSM time series generated by subtracting consecutive DSM pairs from each other.

**L320 ???geomorphological features?**

Yes, we refer to geomorphological destabilization features such as crevasses, cracks, and scarps. We added a clearer definition in the introduction and again in the new methods

subsection on the geomorphological analyses. (Definition in the revised introduction: "In the following, we use the term "destabilization signs" to refer to visible morphological features such as scarps, crevasses, and cracks that develop at the onset of and during destabilization.")

**L334: why? these has to be mentioned in the methods section!**

The data quality of the first five DSMs is not as good as later on due to the quality of the aerial imagery used to derive these DSMs. The DSMs computed from the available digitized analogue images do not include the same level of topographic detail as the DSMs derived from laser scanning, which is an active remote sensing technique. We have restructured the methods section to include this information (Subsection "Long-term change monitoring") and have also added it again here in the results section. The process of deriving DSMs from the aerial imagery is described in previous publications. As part of the overall restructuring of the methods and data section, we have emphasized this more strongly to differentiate between previously published and new data.

**P18 Fixed typo**

**Figure 5 it is easier to mark geomorphological features by a line and not a letter!**

We use the letters to refer to zones of the rock glacier, rather than individual geomorphological features. We would also like the features to remain visible to the reader in the images and would prefer not to mark them with lines, as this would obscure the complexity of the features. For these reasons we would prefer to keep the figure as is, but we will change it if the reviewer or editor feels strongly about this.

**Figure 8: figure is too small and hard to read**

Changed figure layout to improve readability.

**P22 L391 ff what do you mean?**

Rephrased here to improve clarity, additional extended explanation added to the methods section. The image correlation technique is applied at regularly spaced nodes (raster cells) in a distance of 5m. Applying the technique at each node wouldn't be possible from a computational perspective. However, the nodes left out could provide slightly different results compared to the closest considered node. A conceptual figure has been added to the methods section.

**P23 why now named as profiles 0, .. and not P0 ?**

Changed to P0 and checked consistency throughout.

**P25: L439 when?** Timespans I-V in summer 2019, as shown in Fig. 12. We added "In July and August 2019" to the text and refer to Fig 12 for exact dates.

The discussion about the uncertainties in the first part of the chapter is important but unfortunately are the methods and results not presented in the previous chapters. I am missing a structure e.g. chronological order of the datasets

We have restructured and extended the discussion on uncertainties to improve clarity, structure, and completeness. Specifically, we added:

- Overview figure with chronological timeline of data availability to complement the list of data sets in Table 2.
- Additional figure showing uncertainties in North-South, East-West, and vertical direction. This complements another additional figure in the methods section that explains the conceptual basis of the uncertainty analysis.
- Restructured and clearer description of the reasoning for our uncertainty analysis in the text.

P25: Hard to read - Rephrased for clarity.

**P26 L452 what does this mean?**

We added a better explanation as well as a conceptual figure in the restructured methods section and reference this here for clarity.

**L454 but this is not presented in this study.**

We have restructured both the methods section and this section to better describe and explain the co-registration of the multi-temporal datasets. The boxplots originally shown in the supplementary material are now included in the manuscript as an additional and revised figure to aid interpretation.

**L460 showing the uncertainties in m/a makes it difficult to compare the quality of the datasets. Using the absolute value is more suited when discussing the data quality**

In the light of the scope of the analysis - an assessment of velocity changes - we respectfully disagree with this statement. The periods between subsequent epochs are irregular, which would make it more tricky to compare derived distances directly. After normalizing the distances by the time periods, the mean annual velocities and their velocities become directly comparable. We have added explanation of our reasoning to the text and more clearly include references to publications that go into greater detail on absolute uncertainty measures of the previously published data sets.

**L 464 if you discuss this you have to mention it in the method section!**

This refers to work presented in previous studies. We have rephrased the text to clarify this and referenced the relevant publications here and in the methods section.

**P28: a comprehensive table or figure would make it easier to follow the time series**

We have added a figure showing the years for which different kinds of data sets are available in the data and methods section and reference the figure here in the discussion.

P28 add the date: 1953-2006, added to the text.

**P29 how can you know that the onset of destabilization was in 2017 when you do have no data between 2011 and 2016?**

This statement is based on destabilization signs visible in the 2017 DSM. We do have additional annual block displacement data from the 4 block profiles for 2011-2016. We have rephrased the

sentence to make clear what we are referring to and to better express the ambiguity associated with the lack of DSMs between 2011 and 2017. Revised text:

"For HEK, the onset of a second cycle of destabilization can first be definitively observed in the 2017 DSM, after a longer but more gradual acceleration phase. Increases in velocity and strain rates are apparent in the data from in situ block monitoring two to three years prior to 2017, which could suggest earlier destabilization onset. The timing of the onset cannot be determined more exactly due to the gap in DSMs between 2011 and 2017."

**which region?**

The majority is in the French and Swiss Alps, a smaller number from the Austrian and Italian Alps. Added "mostly in the French and Swiss Alps" to the text.

**P31: Subsequent?? - yes, added "subsequent" to the text.**

in the supplement only summer precipitation is shown. What is with snow accumulation? 2017 was little in snow

Added a note on low snow in the 2016/17 winter season to the text and adjusted the figure in the supplement to include solid precipitation per water year.

**???** Fixed typo.

**P32: since about?**

Changed to: "annual resolution since 1997 with a one year gap in 2005 when no measurements were carried out"

**hard to read**

Rephrased and shortened the sentence.

Thank you again for the detailed review and thoughtful comments!

**References**

Besl, P. J. and McKay, N. D.: Method for registration of 3-D shapes, in: Sensor fusion IV: control paradigms and data structures, vol. 1611, pp. 586–606, Spie, 1992.

Conrad, O., Bechtel, B., Bock, M., Dietrich, H., Fischer, E., Gerlitz, L., Wehberg, J., Wichmann, V., and Böhner, J.: System for automated geoscientific analyses (SAGA) v. 2.1. 4, Geoscientific Model Development, 8, 1991–2007, 2015.

Marcer, M., Serrano, C., Brenning, A., Bodin, X., Goetz, J., and Schoeneich, P.: Evaluating the destabilization susceptibility of active rock glaciers in the French Alps, The Cryosphere, 13, 141–155, 2019.

---

## Author Response (AR2)

We thank the editor for her comments and the time spent on the manuscript!
Below are brief responses to the main points raised in the "technical corrections" stage. We fixed the smaller typos and similar issues marked in the pdf as suggested and went through the list of references to add missing bibliographical information.

- P2: Cicoira et al 2020 is the intended citation.
- Fig 1: Adjusted labeling of summits to be consistent across the panels. Changed red lines to not overlap the maps, fixed coordinates. Updated figure caption to include source of base data (SRTM) and explanations of all relevant colours of the permafrost index map.
- P12: rephrased sentence to be more precise. Values were aggregated within the area covered by the block positions as they are shown in Fig 1. A shapefile of these areas is included in the code repositories cited at the end and could be referred to here if needed.
- Fig 6 and 9: Changed "1997-06" label to "1997-2006"
- Fig 8: Made fonts bigger, moved axis labeling of the inset plot to the other side to avoid overlapping lines.
- P23: yes, "phase" here refers to a time interval.
- P34: yes, changed to Switzerland instead of CH.
- P38: yes, good point, changed sentence to reflect this.